# Investigating the volume and diversity of data needed for generalizable antibody–antigen ΔΔG prediction

**Alissa M. Hummer** ✉, **Constantin Schneider, Lewis Chinery & Charlotte M. Deane** ✉

Antibody–antigen binding affinity lies at the heart of therapeutic antibody development: efficacy is guided by specific binding and control of affinity. Here we present Graphinity, an equivariant graph neural network architecture built directly from antibody–antigen structures that achieves test Pearson's correlations of up to 0.87 on experimental change in binding affinity ($\Delta\Delta G$) prediction. However, our model, like previous methods, appears to be overtraining on the few hundred experimental data points available and performance is not robust to train–test cut-offs. To investigate the amount and type of data required to generalizably predict $\Delta\Delta G$, we built synthetic datasets of nearly 1 million FoldX-generated and >20,000 Rosetta Flex ddG-generated $\Delta\Delta G$ values. Our results indicate that there are currently insufficient experimental data to accurately and robustly predict $\Delta\Delta G$, with orders of magnitude more likely needed. Dataset size is not the only consideration; diversity is also an important factor for model predictiveness. These findings provide a lower bound on data requirements to inform future method development and data collection efforts.

Antibodies mediate their functions, both physiologically and therapeutically, by binding specifically to a target antigen. Controlling affinity is therefore the driving consideration in therapeutic antibody development when identifying, as well as optimizing, a lead candidate.

Many other properties, aside from affinity, often referred to collectively as developability, also play important roles. There have been substantial advances in recent years in using machine learning (ML) to predict such properties—from self-association[1] and humanness[2–5] to polyreactivity and specificity[6,7]. However, changes to the antibody sequence to improve these properties must not come at the cost of binding. Thus, therapeutic antibody development relies on solving a complex, multiparameter optimization problem[8,9].

Experimental techniques for affinity quantification are typically slow and laborious[10]. A fast and accurate computational predictor of change in affinity would fill a need in the antibody design pipeline. Furthermore, computational approaches can, in principle, incorporate information from different predictors to simultaneously optimize multiple properties, while still controlling binding affinity.

In silico prediction of antibody–antigen affinity remains a challenge. Traditional affinity prediction tools, such as FoldX[11] and Rosetta Flex ddG[12], are based on physical equations and empirical measurements. These methods recapitulate the main physical properties of the system and have proven effective for certain applications[13] but can be limited in speed and accuracy[12,14]. In recent years, there has been a shift toward ML approaches, which can be divided into two main categories: sequence-based and structure-based. Sequence-based methods have been successfully applied to predict affinity for a specific antigen in cases where a large amount of data is available[15,16]. These methods are not broadly generalizable: the information they are trained on is antigen-specific, and the models cannot be readily applied to another antigen without further training. Structure-based methods promise greater generalizability by aiming to capture the interaction patterns across many different antibody–antigen complexes. Current

Department of Statistics, University of Oxford, Oxford, UK. ✉e-mail: contact@alissahummer.com; deane@stats.ox.ac.uk

methods are trained on features derived from antibody–antigen complex structures, such as binding surface area, interatomic interactions and energy-based terms[14,17,18]. However, these methods appear to not predict well outside their training data[17,19]. In addition, they require the extraction of features, which can be slow and is subject to human bias.

Here, we investigate the ability of and requirements for ML methods to predict changes in antibody–antigen binding affinity ($\Delta\Delta G$). We developed an equivariant graph neural network (EGNN) architecture, Graphinity, which achieved Pearson's correlations of up to 0.87 on the AB-Bind dataset[20] of 645 single-point mutations. However, further investigation indicated that this high performance stemmed from model overtraining and was not generalizable, an observation that has been found for previous approaches[17,19,21,22].

To explore the volume and type of data that would be needed to build accurate methods, we generated synthetic datasets of nearly 1 million FoldX[11] and over 20,000 Rosetta Flex ddG[12] $\Delta\Delta G$ values. On the large FoldX dataset, Graphinity achieved Pearson's correlations close to 0.9, which were robust to train–test sequence identity cutoffs and noise.

Assessing model performance with varying amounts of synthetic data demonstrated that there are currently insufficient experimental data for generalizable $\Delta\Delta G$ prediction and that orders of magnitude more are likely needed. Our results set a lower bound on the amount of data required and highlight the importance of dataset diversity for model predictiveness.

## Results

### Graphinity model architecture

Graphinity takes structures of a wild-type (WT) and a mutant antibody–antigen complex as input, feeds the corresponding graph representations through a Siamese EGNN[23] and predicts $\Delta\Delta G$ (Fig. 1a). In the atomistic graphs, non-hydrogen atoms are represented as nodes and interactions between nodes less than 4 Å apart as edges. Graphs are limited to the neighborhood around the mutated site. The architecture is modular and easily adapted for regression and classification and for single- and multi-graph inputs (see 'Graphinity: EGNN architecture' section in the Methods for full details).

### Graphinity performance for predicting experimental $\Delta\Delta G$

We applied Graphinity to the experimental $\Delta\Delta G$ dataset from AB-Bind[20], which contains 645 single-point mutations from 29 complexes and will from here on be referred to as Experimental_$\Delta\Delta G$_645 (Supplementary Table 1 and Supplementary Fig. 1a; see Fig. 1c for an example of $\Delta\Delta G$ data). We considered hypothetical reverse mutations (Experimental_$\Delta\Delta G$_645 ± Reverse Mutations), as well as non-binder mutations in the AB-Bind dataset whose $\Delta\Delta G$ values were arbitrarily set to −8 kcal mol⁻¹ (Experimental_$\Delta\Delta G$_645 ± Non-Binders).

Our model achieved Pearson's correlations of up to 0.87 on 10-fold cross-validation (Fig. 2a), outperforming existing methods that report correlations of up to 0.76 (refs. 17,18). However, delving into the robustness of the model—by implementing sequence identity cutoffs between folds—indicated that these high correlations were the result of overtraining as opposed to true learning (Fig. 2b). When we imposed a 100% length-matched complementarity-determining region (CDR) sequence identity cutoff, ensuring that mutations from the same complex cannot be in both the training and test datasets, the Pearson's correlations decreased by an average of 63%. The results were also highly sensitive to the inclusion of non-binders (Fig. 2b) and, for all train–test cutoffs, there was substantial variation in the Pearson's correlation across different folds (Supplementary Fig. 2).

Limitations in model generalizability for experimental $\Delta\Delta G$ prediction have been found for previous approaches when train–test cutoffs were imposed[17–19,21,22]. For example, a leave-one-complex-out test of TopNetTree caused a drop in the average Pearson's correlation to 0.17 (ref. 17). DGCddG[24] and RDE-PPI Network[25], general protein–protein interaction $\Delta\Delta G$ prediction methods that we were able to retrain on the

Experimental_$\Delta\Delta G$_645 dataset (90% CDR sequence identity cutoff), achieved correlations of 0.26 and 0.22, respectively (Supplementary Table 2).

Results on a benchmark dataset we generated from the SKEMPI 2.0 database (Experimental_$\Delta\Delta G$_608), which includes a larger number of antibody–antigen complexes, show similar trends (Supplementary Results, 'SKEMPI 2.0 benchmark dataset'; Supplementary Fig. 3).

### Using a synthetic dataset of ~1 million mutations

The poor robustness of model performance on the limited experimental data led us to investigate how well $\Delta\Delta G$ could be predicted if more data were available. We explored the use of FoldX[11] and Rosetta Flex ddG[12] to create larger synthetic $\Delta\Delta G$ datasets (Supplementary Table 1 and Supplementary Figs. 1c,d and 4). Computational costs limited the number of mutations that could be modeled with Flex ddG (Supplementary Table 3), and therefore most further analysis will focus on the FoldX dataset.

We generated nearly 1 million $\Delta\Delta G$ data points (Synthetic_FoldX_$\Delta\Delta G$_942723; Supplementary Table 1 and Supplementary Fig. 1c) by exhaustively mutating the interfaces of structurally resolved complexes from the Structural Antibody Database (SAbDab)[26,27] using FoldX (Fig. 1b). FoldX uses physical equations and empirical measurements to generate predictions of binding affinity. The resulting synthetic dataset will not completely mimic the complexity of true $\Delta\Delta G$ values, but FoldX captures the key features underlying molecular interactions. The Pearson's correlation between FoldX predictions and experimental values is 0.34 for the AB-Bind dataset[20]. The accuracy is higher for mutations with a larger effect on binding affinity though. The area under the receiver operating characteristic curve (ROC AUC) in predicting whether a mutation is stabilizing or not is 0.87 for mutations with an absolute value greater than 1 kcal mol⁻¹ (ref. 20), supporting that these data contain some of the characteristics of experimental values.

On this synthetic dataset, Graphinity achieved a test Pearson's correlation of 0.89 with 10-fold cross-validation and a 90% length-matched CDR sequence identity cutoff imposed between folds (Fig. 2c). Training the model for longer (100 epochs, as opposed to 10) improved the correlation slightly, to 0.92 on a single fold, but we did not explore this further as it was computationally costly to run (see 'Graphinity: EGNN architecture' section in the Methods).

Graphinity outperformed other approaches for predicting $\Delta\Delta G$ on this dataset. A simple baseline, the change in number of contacts between the WT and mutant structure (4 Å interaction distance cutoff), achieved a correlation of 0.42 with the synthetic $\Delta\Delta G$ values (full dataset). We also tested multiple ML methods (Fast Library for Automated Machine Learning (FLAML)[28], convolutional neural network (CNN)[15], Rotamer Density Estimate (RDE)[25] and Equiformer[29,30]) and inputs (sequence- and structure-based) on a held-out test set (Supplementary Table 4; see 'Structure-informed sequence-based models', 'ESM2 embedding-based model', 'Equiformer' and 'Comparison against protein–protein interaction $\Delta\Delta G$ prediction methods' sections in the Methods). The graph-based approaches achieved the strongest performance (correlations of 0.87 and 0.89 for the EGNN and Equiformer architectures, respectively). As the EGNN required substantially less memory and time to train, subsequent analyses were conducted with this architecture.

The performance of Graphinity was robust to a range of train–validation–test sequence identity cutoffs (Figs. 2b and 4a). The most stringent split, a length-matched CDR sequence identity cutoff of 70% plus an antigen sequence identity cutoff of 70%, maintained a Pearson's correlation of 0.89. However, consistent with known difficulties in predicting out-of-domain, the model correlation on train–validation–test data split on the basis of affinity was only 0.52 (Supplementary Fig. 5; Supplementary Results, 'Affinity-based train–validation–test split').

Another way to assess model performance is with the Spearman's rank correlation. This value (~0.64) was lower than the Pearson's

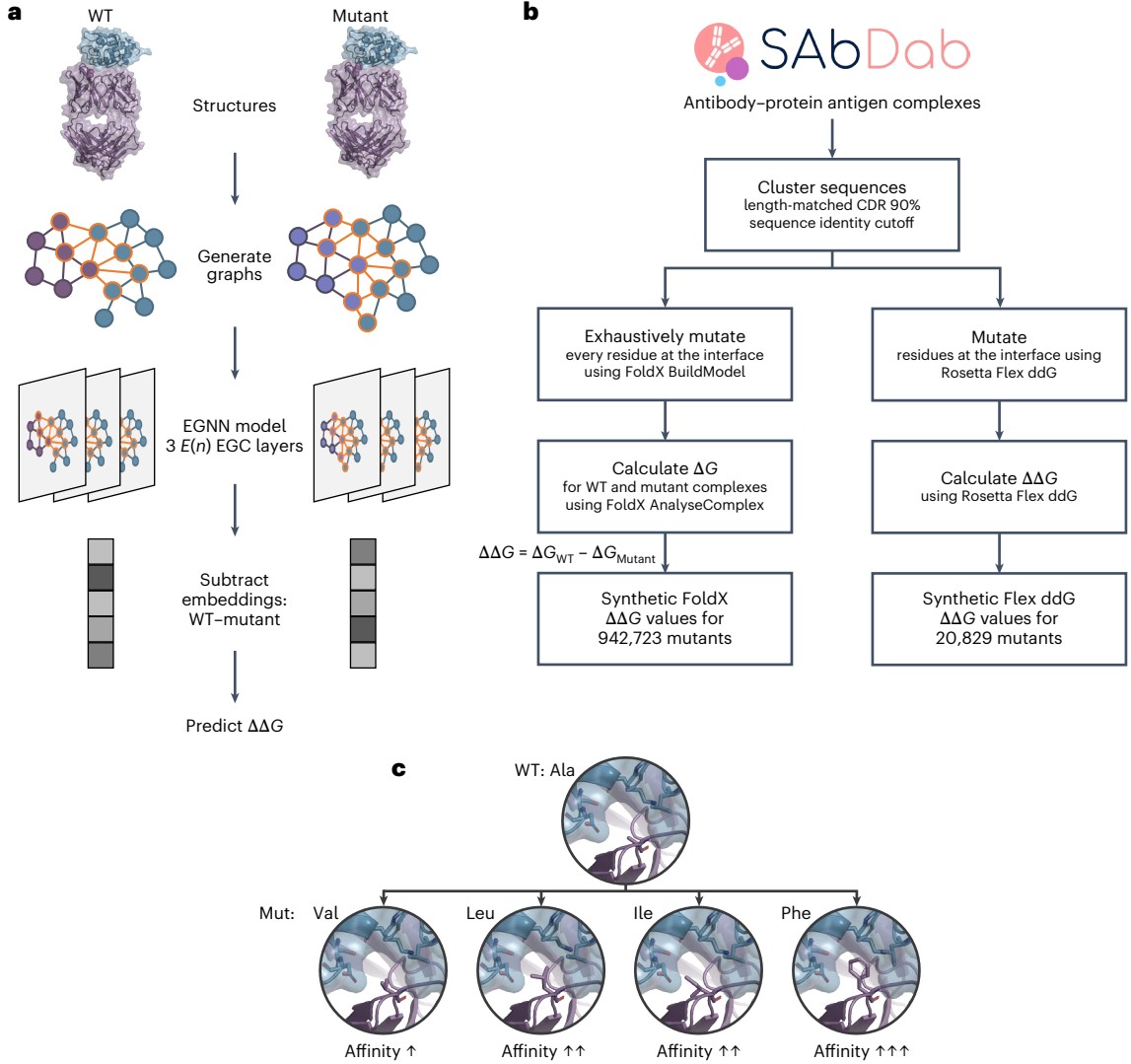

**Fig. 1 | Graphinity architecture and synthetic dataset preparation. a,** The EGNN deep learning models are trained on graphs of three-dimensional protein structure coordinates. The graphs are built from atoms in the neighborhood of the mutated site, conceptually illustrated with circles (nodes, atoms) and connecting lines (edges, interactions). The antibody is shown in purple, the antigen in blue and inter-binding partner edges in orange. Our model architecture consists of three $E(n)$ EGC layers[23], followed by a linear layer. For $\Delta\Delta G$ prediction, the embeddings generated from the $E(n)$ EGC layers for the WT and mutant complex are subtracted from one another before passing through the linear layer. **b,** The synthetic $\Delta\Delta G$ datasets were generated from structurally resolved complexes from SAbDab[26,27]. We mutated interface residues and predicted $\Delta\Delta G$ values using FoldX[11] and Rosetta Flex ddG[12]. **c,** An example of $\Delta\Delta G$ data for a complex. PDB: 1XGP[56]; antibody in purple, antigen in blue; affinity values from SKEMPI 2.0[31]. Mut, mutant.

correlation for our models. This appears to be due, in large part, to the high density of $\Delta\Delta G$ values close to 0, which the EGNN did not always rank correctly. The Spearman's rank correlation rose to ~0.74 when values between −1 and +1 kcal mol⁻¹ were excluded. FoldX is known to be less accurate at predicting the $\Delta\Delta G$ values for mutations with only a small effect on binding affinity[20], and therefore there may be less signal in the data in this region.

In addition, the FoldX dataset includes values for highly disruptive mutations that fall below the range of $\Delta\Delta G$ values that can currently be measured accurately experimentally. While the exact cutoff depends on the sensitivity of the assay and the comparative WT $\Delta G$ value, mutations with a $\Delta\Delta G$ value less than −12.2 kcal mol⁻¹ (the lowest value in the SKEMPI 2.0 database[31]) are likely to be non-binding. Limiting the test dataset to mutations with a FoldX $\Delta\Delta G > −12.2$ kcal mol⁻¹ (99% of the total values) resulted in a slightly lower test Pearson's correlation of 0.78.

We also investigated model performance with different graph inputs. On the full interface rather than just the mutation site

neighborhood, reflecting the input for potential multi-point mutation data, performance was maintained (Pearson's correlation of 0.87). In a preliminary analysis with predicted structure inputs, we applied Graphinity (without further training) to a dataset of 100 randomly selected mutations whose structures we modeled with Boltz-1[32] (see 'Testing Graphinity on modeled structure inputs' section in the Methods). Graphinity was not predictive on this dataset (Pearson's correlation of 0.02), consistent with the remaining challenges in modeling antibody–antigen complexes[33].

The successful application of Graphinity to a large synthetic dataset serves as a proof of concept that $\Delta\Delta G$ can be accurately predicted when sufficient data are available.

### Dataset size in experimental $\Delta\Delta G$ dataset generation

Having demonstrated the potential of the EGNN architecture for predicting $\Delta\Delta G$ when training data are abundant, we next attempted to quantify the amount of data required for the accurate prediction of experimental values. We built models with

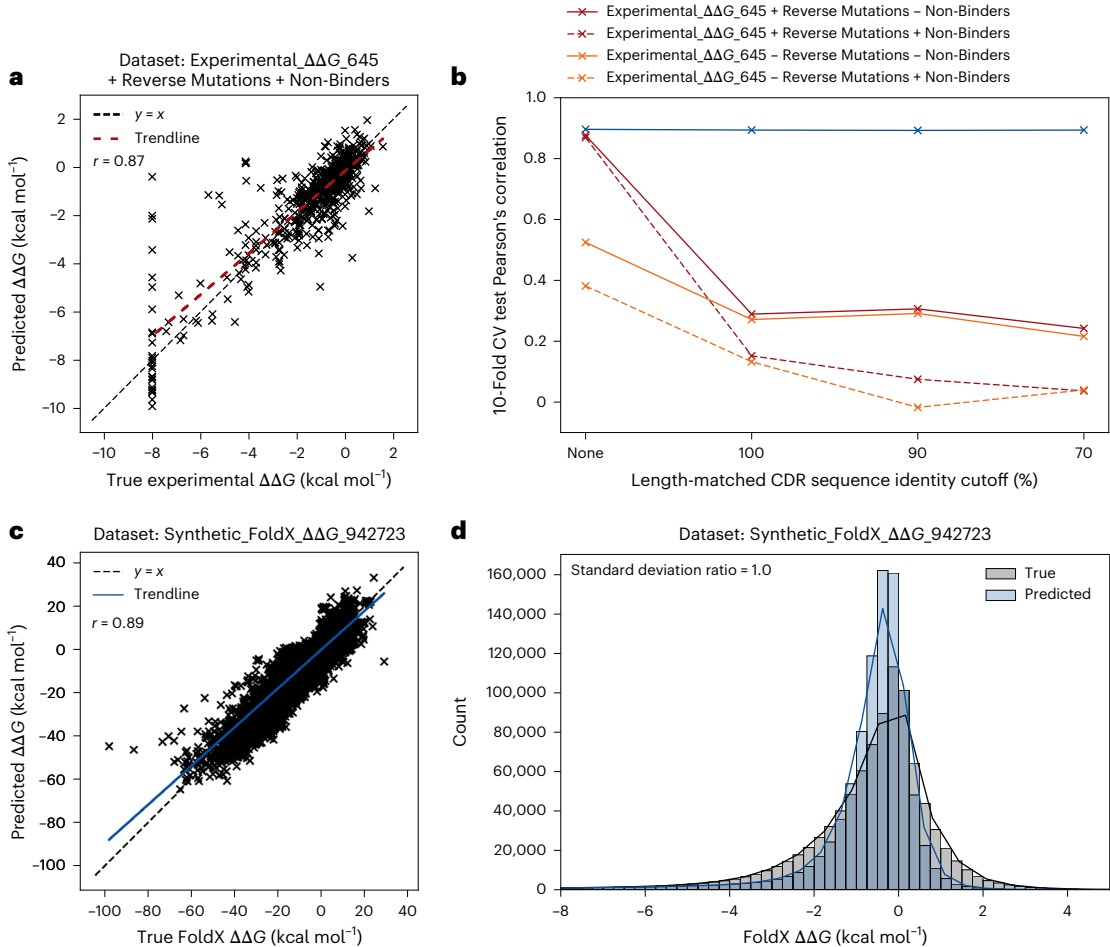

**Fig. 2 | Graphinity model performance for ΔΔG prediction. a**, Correlation of Graphinity predictions with true experimental values for the Experimental_ΔΔG_645 + Reverse Mutations + Non-Binders dataset, with a random train–validation–test split. Reverse mutations were used for training and validation only and were not included in the test dataset. An ensemble of ten models was trained for 500 epochs with 10-fold cross-validation (CV) on the datasets. The trendline, shown in red, is a least-squares polynomial fit. **b**, The effect of train–validation–test CDR sequence identity cutoffs on Graphinity performance. This figure is included with error bars representing the standard deviation across the

ten folds in Supplementary Fig. 2. **c**, Correlation of Graphinity predictions with true synthetic values for the Synthetic_FoldX_ΔΔG_942723 dataset with a 90% length-matched CDR sequence identity cutoff applied for the train–validation–test split. An ensemble of ten models was trained for ten epochs with 10-fold cross-validation. The trendline, shown in blue, is a least-squares polynomial fit. **d**, Histograms of the true and predicted FoldX ΔΔG values shown in **c** (x axis limited to −8 to +5 kcal mol⁻¹ for clarity). The solid lines are kernel density estimates (KDEs). The Pearson's correlation (r) values are shown in **a** and **c**.

varying training plus validation dataset sizes (datasets Synthetic_FoldX_ΔΔG_{580-450000}; Supplementary Table 1) and applied them to a test set of 94,126 mutations (90% length-matched CDR sequence identity cutoff). Test Pearson's correlations only began to plateau, reaching 0.85, for models trained with at least 90,000 mutations (Fig. 3a).

Upon comparing the distributions of the predicted and true values, we observed that the models built from smaller datasets often regressed toward the mean and achieved high correlations despite predictions not covering the full range of true values. To quantify this effect, we calculated the standard deviation ratio, the relative ratio of the standard deviations of the true and predicted values (lower divided by higher value, standard deviation ratio <1). The standard deviation ratio only exceeded 0.8 with a dataset size of 450,000 mutations (Fig. 3a).

As any estimates of data requirements will be influenced by the ML model and the nature of the synthetic dataset, we evaluated our predictions with further approaches. We applied the Equiformer architecture to the varying training data subsets and found a similar trend in performance as was observed for the EGNN (Supplementary Fig. 6).

Furthermore, we generated another synthetic dataset using Rosetta Flex ddG[12]. Flex ddG takes substantially longer to run per mutation (Supplementary Table 3), constraining the number of mutations feasible to model, but offers an alternative physics-based parameterized method to explore model performance and data requirements. Flex ddG achieved a higher correlation to experimental values than FoldX, although both methods recapitulate the physical properties of protein–protein interactions and both are limited in accuracy (Supplementary Fig. 4). On a dataset of 20,829 Flex ddG mutations (Synthetic_FlexddG_ΔΔG_20829; Supplementary Table 1 and Supplementary Fig. 1d), we observed slightly lower performance but overall similar trends as on the FoldX dataset (Supplementary Results, 'Synthetic FlexddG dataset'; Supplementary Fig. 7). These results support that orders of magnitude more experimental data will be required to achieve accurate and generalizable prediction of ΔΔG.

### Dataset diversity in experimental ΔΔG dataset generation

Diversity is a known important characteristic of any dataset used for model training. We evaluated the role of dataset diversity using three metrics: the diversity of antibody sequences, amino acid substitution

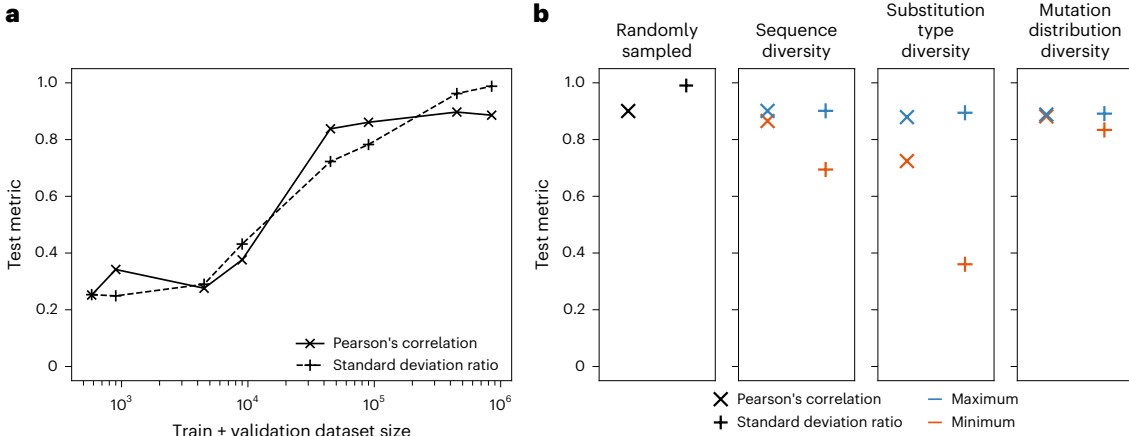

**Fig. 3 | Considerations for experimental ΔΔG dataset generation, with respect to ML predictiveness. a,b,** Model performance with varying training plus validation dataset size (datasets: Synthetic_FoldX_ΔΔG_{580-450000}, Supplementary Table 1) (**a**) and dataset diversity (datasets: Synthetic_FoldX_ΔΔG_100000_randomly_ sampled, Synthetic_FoldX_ΔΔG_100000_{sequence/substitution_type/ substitution_distribution}_{min/max}; Supplementary Table 1) (**b**). For **b**, we considered diversity in antibody CDR sequence identity, amino acid substitution type frequency and the distribution of mutated positions in the complex.

types and structural distribution of mutations in the interface (for more details, see 'Varying synthetic dataset diversity' section in the Methods). We constructed training and validation datasets to minimize and maximize each respective metric in the FoldX datasets (Synthetic_FoldX_ΔΔG_100000_{sequence/substitution_type/substitution_distribution}_{min/max}; Supplementary Table 1). For example, the Synthetic_FoldX_ΔΔG_100000_sequence_min training dataset contained mutations from 75 antibody–antigen complexes, while the corresponding maximum-diversity dataset contained mutations from 1,177 complexes. All models built from these datasets were evaluated on the same test data, consisting of 10,000 mutations (Supplementary Table 1; 90% length-matched CDR sequence identity cutoff).

The distribution of mutations in the interface had only a marginal effect, which may be explained by the input graphs, which represent only the neighborhood of the mutated site. However, we found that sequence and substitution type diversity impacted model performance, particularly the test standard deviation ratio (Fig. 3b). The minimum sequence and substitution type diversity datasets achieved 23% and 60% lower standard deviation ratios than the corresponding maximum diversity datasets, respectively.

### Impact of noise on a large synthetic ΔΔG dataset

Experimental ΔΔG data are noisy, particularly if acquired from different experimental setups and/or laboratories[31]. We explored the robustness of Graphinity to noise by perturbing the training and validation datasets of Synthetic_FoldX_ΔΔG_942723 in two ways: (1) shuffling the affinity labels corresponding with mutations (Synthetic_FoldX_ΔΔG_942723_ shuffled) and (2) adding Gaussian-distributed random noise to the labels (Synthetic_FoldX_ΔΔG_942723_gaussian_noise).

The Pearson's correlations on held-out test sets remained remarkably constant, at approximately 0.85 for datasets with 0–60% shuffled labels (Fig. 4b). However, upon analyzing the relative distributions of the predicted and true FoldX ΔΔG values, we found that the model lost predictiveness with increased shuffling: the predicted values began to fall in increasingly narrow distributions (Fig. 4b). Model performance was 0 when 100% of the labels were shuffled, supporting that, while the FoldX-generated values are not as accurate as experimental data, there is true signal that can be learned from the input complex structures.

There are 82 duplicated antibody–antigen single-point mutations in SKEMPI 2.0[11] that did not result in non-binders or have imprecisely measured affinity. Across these, the average ΔΔG standard deviation between duplicates is 0.19 kcal mol⁻¹ and the maximum is

0.90 kcal mol⁻¹. Graphinity maintained Pearson's correlations and standard deviation ratios above 0.8 with added noise in this range, and indeed up to a Gaussian noise scale of 5 (Fig. 4c).

### Performance by amino acid substitution

We further investigated how our model performs for specific amino acid substitutions (for example, Arg to Lys). The overall pattern of the mean ΔΔG values and corresponding standard deviations closely matched between the experimental and predicted values (Supplementary Fig. 8), suggesting that the model learns structural context. The Pearson's correlation between the average ΔΔG values for specific substitutions and the true values is just 0.35, as compared with the trained model's performance of 0.89.

The FoldX ΔΔG values varied widely for a specific substitution, with standard deviations ranging from 0.5 to 10.6 kcal mol⁻¹ (Supplementary Fig. 8c). Certain mutations (for example, mutations from Gly or mutations to Phe, His, Trp or Tyr) exhibited noticeably lower average ΔΔG values and higher standard deviations, consistent with the disruptive effects that can occur from replacing a smaller with a larger amino acid. The EGNN model achieved higher performance on these mutations (Supplementary Fig. 9; Supplementary Results, 'Performance by amino acid substitution'). Future experimental data generation should enrich for mutations that proved more challenging to predict (for example, mutations to small residues).

### Testing Graphinity on an experimental binding dataset

To test whether Graphinity can learn the distribution of experimental data, we adapted and applied our architecture to a dataset of 36,391 CDRH3 variants of trastuzumab[15] (for details, see 'Trastuzumab variants' section in the Methods). The variants are classified as binders or non-binders for the antigen, HER2. This task, where binding is considered only for a single antigen, is simpler and not necessarily the intended aim of the Graphinity architecture. However, this dataset was sufficiently large that we would expect prediction to be successful.

Our model learned to separate the binding and non-binding variants, achieving a ROC AUC of 0.90 and average precision (AP) of 0.82 (Supplementary Fig. 10a). This performance is close to that of the sequence-based CNN previously applied to this dataset (ROC AUC of 0.91, AP of 0.83)[15]. Furthermore, our performance was robust to train–validation–test cutoffs with ROC AUC values maintained above 0.90 when V- and J-gene clonotype plus CDRH3 sequence identity cutoffs (90% and 70%, respectively) were applied (Supplementary Fig. 10b).

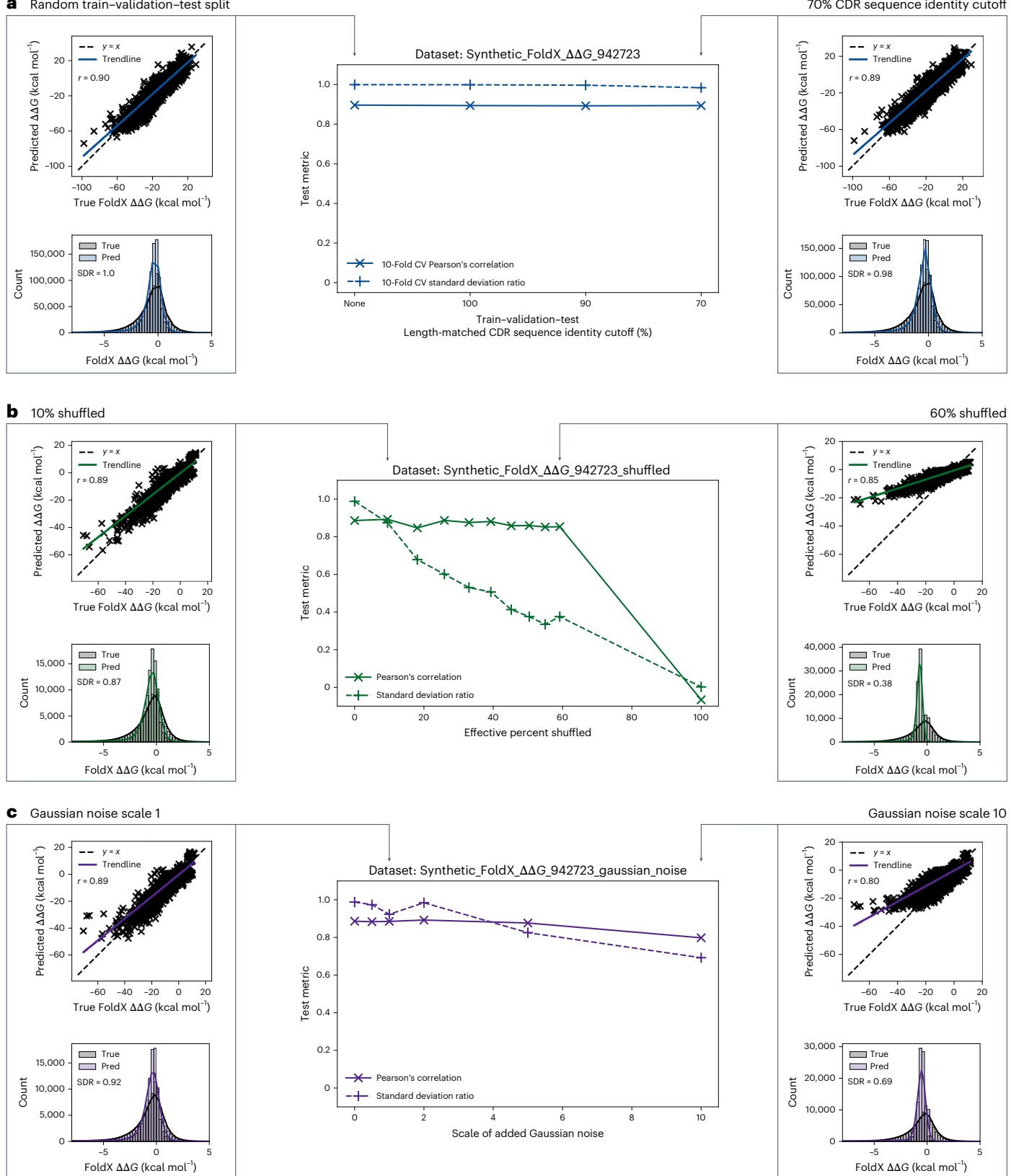

**Fig. 4 | Model robustness to train–validation–test cutoffs and noise.**
**a–c**, The influence of train–validation–test cutoffs (**a**), shuffling (Synthetic_
FoldX_ΔΔG_942723_shuffled) (**b**) and Gaussian noise (Synthetic_FoldX_
ΔΔG_942723_gaussian_noise) (**c**) on model performance. Results are shown
for 10-fold cross-validation in **a** and for a single fold, held-out test set in **b** and

**c**. The Pearson's correlation (*r*) values are shown in the scatter plots in **a**–**c**. For
all histograms, the *x* axes were limited to −8 to +5 kcal mol⁻¹ for clarity and the
solid lines are kernel density estimates (KDEs). Pred, predicted; SDR, standard
deviation ratio.

## Discussion

Antigen binding affinity, essential to the function and efficacy of an antibody, is complex and challenging to predict computationally. ML models have achieved strong performance when trained on small experimental datasets, such as AB-Bind[20], with random train–test splits. However, these high correlations are the result of overfitting, and the performance does not generalize to complexes dissimilar from the training data[17,19,21,22]. To prevent information leakage, effective sequence identity cutoffs between train and test datasets are essential.

To test whether affinity could be accurately and robustly predicted if more data were available, we applied ML to a synthetic dataset of nearly 1 million mutants generated using FoldX[11]. Assessing different architectures based on sequence and structure inputs validated the suitability of graph-based deep learning for ΔΔG prediction. The Graphinity EGNN model achieved high correlations that were maintained under stringent train–test sequence identity cutoffs for both the antibody and antigen, as well as the levels of noise observed in experimental datasets.

Our results on the synthetic values must be considered in light of the source of the data. The data points were all produced by the same software and are thus expected to be more self-consistent and less noisy than experimental data. The synthetic values may also follow a different distribution to the true values. However, FoldX can accurately predict whether mutations with a substantial effect on binding affinity will be stabilizing or destabilizing, suggesting there is signal in this dataset[20].

We tested that the Graphinity architecture can learn the distribution of experimental data, not just synthetic values, by applying it to an experimental dataset of >36,000 trastuzumab variants. The EGNN model achieved strong performance similar to a previous CNN, but offers further benefits, most notably the potential for generalizability to different antibody–antigen complexes.

The success of ML on large datasets lends support to the idea that the major challenge with experimental ΔΔG prediction lies in data availability rather than model architectures. We explored the amount of data that would be required for accurate and generalizable prediction of experimental ΔΔG using the synthetic data. Our results with the EGNN and Equiformer architectures and on the FoldX and Flex ddG datasets suggest that there are currently vastly insufficient experimental data available and orders of magnitude more, tens to hundreds of thousands of data points, will likely be needed. Under the conditions tested in this study, we estimate that at least 90,000 ΔΔG values must be obtained to achieve test Pearson's correlations exceeding 0.85. Such a volume of data will become more attainable with increases in the throughputs of experimental methods (for example, ref. 34). The exact required dataset size is, however, dependent on dataset diversity and the ML method, and liable to change in the future. As more data become available, there is greater potential for limitations in data to be compensated for, to some extent, by ML know-how such as by identifying model architectures that require less data, using stratified sampling, or by transfer learning from a related data-rich task or from synthetic data. Future model design could be augmented by considering physiological features that are typically ignored in current methods, such as water molecules and protein conformational flexibility.

In addition to dataset size, we find that dataset diversity, particularly with respect to antibody sequence identity and amino acid substitution type, is an important factor. Both of these diversity metrics are currently very limited in experimental data. For example, the antibody–antigen single-point mutations in SKEMPI 2.0[31] derive from fewer than 50 complexes and are highly skewed in substitution type, with mutations to alanine making up over half of the dataset.

Our results highlight the need to move toward 'machine learning-grade' data, where model development informs the data generation process. In addition, several key efforts should be considered to advance generalizable affinity prediction: (1) increasing the throughput of experimental methods for affinity measurement (for example, ref. 34), (2) designing diverse and well-structured datasets, (3) establishing a standardized, regularly updated repository for ΔΔG values and metadata, and (4) implementing robust, blind assessments of model performance (for example, ref. 35). These measures would ultimately improve the reliability of ML for antibody–antigen affinity prediction.

## Methods

### Experimental ΔΔG data preparation

The AB-Bind dataset consists of 645 single-point mutations and ΔΔG measurements from 29 complexes. We downloaded this dataset, which was originally compiled by Sirin et al.[20], from TopNetTree[17]. We reversed the sign on the ΔΔG labels to reflect $\Delta\Delta G = \Delta G_{WT} - \Delta G_{Mutant}$, as is done in ref. 18 and our synthetic datasets. We 'repaired' the structures using FoldX (version 5) RepairPDB and modeled the mutations using FoldX BuildModel[11]. We refer to this dataset as Experimental_ΔΔG_645 (Supplementary Table 1).

As in mCSM-AB2[18], we generated reverse mutations by mutating the forward mutant model back to WT using FoldX BuildModel and setting the ΔΔG label to the negative value of the forward mutation (Experimental_ΔΔG_645 + reverse mutations).

The Experimental_ΔΔG_645 dataset has multiple limitations: 5 of the complexes do not contain an antibody (Protein Data Bank (PDB) identifiers: 1AK4, 1FFW, 1JTG, 1KTZ and 3K2M), 27 of the mutations are non-binders whose change in binding affinity has arbitrarily been set to −8 kcal mol⁻¹, and there are 3 duplicated mutations with different ΔΔG values. We have used it here to compare against the performance of previous methods, which were applied to this dataset[14,17,18].

These limitations prompted us to propose an experimental antibody–antigen ΔΔG benchmarking dataset, Experimental_ΔΔG_608 (Supplementary Table 1), consisting of 608 single-point mutations, obtained by rigorous filtering of the SKEMPI 2.0 database[31].

We downloaded the SKEMPI 2.0 database, which contains data on changes to the binding affinity of structurally resolved protein–protein interactions in response to mutations[31]. The total database consists of 7,085 entries, 1,150 of which are for antibody–antigen interactions. We removed non-antibody–antigen complexes and further filtered the antibody–antigen dataset by removing multipoint mutations, non-binder mutations, mutations with imprecisely measured affinity, and duplicates of mutations. When removing duplicate mutations, we preferentially retained those with kinetic data, based on the measurement method (SPR > ITC > KinExA > FL > IASP > SP > CSPRIA > ELISA > BI) and based on the temperature (298 > 296 > 303 > 310 > 283 > 298 (assumed)). The final filtered dataset contained 608 single-point mutations from 44 complexes.

We calculated the binding affinity (ΔG) and change in binding affinity (ΔΔG) values from the SKEMPI 2.0 database as

$$\Delta G = RT \times \ln(K_d), \tag{1}$$

where $R$ is the ideal gas constant, $T$ is the temperature and $K_d$ is the dissociation constant; and

$$\Delta\Delta G = \Delta G_{WT} - \Delta G_{Mutant}, \tag{2}$$

where $\Delta G_{WT}$ and $\Delta G_{Mutant}$ are the binding affinity values of the WT and mutant complex, respectively.

### Synthetic ΔΔG data preparation

To investigate affinity prediction without the constraint of dataset size, we generated two synthetic datasets orders of magnitude larger than the experimental datasets (Fig. 1b). These were generated using physics-based methods, FoldX[11] and Rosetta Flex ddG[12], to model and predict the ΔΔG values of mutations.

We downloaded structurally resolved antibody–protein antigen complexes from SAbDab[26,27], resulting in 6,077 nonredundant

entries from 3,065 PDB files (SAbDab accession date: 19 May 2022). Twenty-seven PDBs with only $C_\alpha$ residues resolved were removed from the dataset. We renumbered the PDB files using a custom script, to prevent issues with insertion numbering in subsequent steps with FoldX, and repaired the PDBs using FoldX RepairPDB[11]. We removed from the dataset 25 PDBs for which the repair did not run to completion. We then clustered the antibody–antigen complexes based on a 90% length-matched CDR sequence identity threshold (see below), resulting in 1,475 clusters.

**FoldX.** One complex per cluster was carried forward for exhaustive interface mutagenesis with FoldX: all interface residues, defined as being within 4 Å of the binding partner, were mutated to every other amino acid using FoldX BuildModel[11]. The FoldX $\Delta\Delta G$ was determined as

$$\Delta\Delta G = \text{InteractionEnergy}_{WT} - \text{InteractionEnergy}_{Mutant}, \quad (3)$$

where $\text{InteractionEnergy}_{WT}$ and $\text{InteractionEnergy}_{Mutant}$ represent the estimated binding affinity values of the WT and mutant complex, respectively, generated from FoldX AnalyseComplex[11]. A negative $\Delta\Delta G$ value represents a destabilizing mutation.

We excluded mutations where the WT amino acid was 'X', the chain identifier was a number, the antibody and antigen were >4 Å apart and the FoldX interaction energy calculation failed. The final dataset (Synthetic_FoldX_$\Delta\Delta G$_942723; Supplementary Table 1) consisted of 942,723 mutations from 1,471 antibody–antigen complexes from 1,409 PDBs.

**Flex ddG.** Rosetta Flex ddG[12] is another method for computational $\Delta\Delta G$ prediction that is considered to be more accurate than FoldX but takes substantially longer to run (Supplementary Table 3). We therefore generated a smaller synthetic dataset with this tool. In addition, to reduce the runtime of Flex ddG, we set the 'nstruct' parameter (number of structures to model) to 1, instead of the default value of 35. The 'backrub_trajectory_stride' was set to 35,000, and all other parameters were set to default values (max_minimization_iter = 5,000, abs_score_convergence_thresh = 1.0, number_backrub_trials = 3,500). The 2020.08+release.cb1caba version of Rosetta was used to run Flex ddG. We benchmarked the performance of Flex ddG under these conditions on the Experimental_$\Delta\Delta G$_608 dataset, derived from the SKEMPI 2.0 database[31]. The Flex ddG values (Talaris 2014 energy function with generalized additive model (GAM) reweighting ('ddG_fa_talaris2014-gam')) achieved a Pearson's correlation of 0.46 with the experimental values (as compared with a correlation of 0.42 for Flex ddG Talaris 2014 without GAM reweighting ('ddG_fa_talaris2014') and 0.20 for FoldX) (Supplementary Fig. 4). The values with GAM reweighting were used. There is a moderate correlation of 0.39 between Flex ddG and FoldX $\Delta\Delta G$ values.

We sampled a subset of the full FoldX synthetic dataset (Synthetic_FoldX_$\Delta\Delta G$_942723), randomly selecting 16 mutations per complex, to model with Flex ddG. Mutations for which Flex ddG failed to produce a modeled structure of the mutant complex were excluded. The signs on the $\Delta\Delta G$ values were reversed to reflect $\Delta\Delta G = \Delta G_{WT} - \Delta G_{Mutant}$, as is done our FoldX dataset. The final dataset (Synthetic_FlexddG_$\Delta\Delta G$_20829; Supplementary Table 1) consisted of 20,829 mutations from 1,302 complexes.

The dataset and download links are available via GitHub at https://github.com/oxpig/Graphinity. The runtimes of FoldX and Flex ddG are given in Supplementary Table 3.

### Train–validation–test cutoffs
We numbered all antibody sequences with ANARCI[36] using the IMGT numbering scheme[37]. The CDRs were extracted, concatenated and binned on the basis of length. We applied CD-HIT[38], with varying sequence identity cutoffs, to cluster the length-matched CDRs. Seventy percent was the lowest threshold rounded to ten for which CD-HIT ran (for CDRs and antigen sequences).

The AB-Bind Experimental_$\Delta\Delta G$_645 dataset contained non-antibody–antigen complexes, which could not be clustered by CDR sequence identity. The sequences of each of the chains in these complexes had less than 90% sequence identity with each other and each complex was considered as its own cluster.

We generated a synthetic FoldX dataset imposing an antigen sequence identity cutoff in addition to the antibody CDR sequence identity cutoff. In this case, antigen sequences were extracted from the PDB structures using the Bio.PDB.PDBParser module and clustered using CD-HIT with a 70% sequence identity cutoff. Clusters from the antibody CDR- and antigen-based sequence identity cutoffs were merged such that no cluster had a complex with >70% length-matched CDR sequence identity to an antibody in another cluster nor >70% sequence identity to an antigen in another cluster.

Train–validation–test datasets were generated with an 80%–10%–10% split, with respect to the full dataset size. The datasets were sampled such that no cluster had members in more than one dataset, with the exception of datasets split with no cutoff. For 10-fold cross-validation, we generated ten dataset folds using the CD-HIT clusters, such that no cluster had members in more than one fold.

The synthetic FoldX dataset was already filtered with a 90% CDR sequence identity cutoff, and as such, the 100% and 90% cutoffs are functionally identical (although these were sampled from the full dataset separately).

Unless otherwise specified, models were built from a single-fold 80%–10%–10% split with a 90% length-matched CDR sequence identity cutoff.

In addition to splitting the train, validation and test data on the basis of sequence identity, we generated a split of the Synthetic_FoldX_$\Delta\Delta G$_942723 dataset based on the $\Delta\Delta G$ values. In this case, the 80% of the dataset with the lowest $\Delta\Delta G$ values ($\Delta\Delta G < 0.32183$ kcal mol$^{-1}$) were assigned to the training dataset, the next 10% (0.32183 kcal mol$^{-1}$ ≥ $\Delta\Delta G$ > 0.802 kcal mol$^{-1}$) to the validation set and the remaining (top) 10% ($\Delta\Delta G$ ≥ 0.802 kcal mol$^{-1}$) to the test dataset.

### Varying synthetic dataset amounts
To investigate the role of dataset size in model performance, we trained models on a subset of the full, large-scale synthetic FoldX dataset (Synthetic_FoldX_$\Delta\Delta G$_942723). These subsets were randomly sampled from the respective train and validation datasets (Synthetic_FoldX_$\Delta\Delta G$_{580-450000}; Supplementary Table 1). All models were evaluated on the same test set, consisting of 94,126 mutations (one fold, held-out test set). A 90% length-matched CDR sequence identity cutoff was applied between respective train, validation and test sets.

Subsets of the synthetic Flex ddG dataset (Synthetic_FlexddG_$\Delta\Delta G$_20829) were also generated and evaluated in the same manner (Synthetic_FlexddG_$\Delta\Delta G$_{580-9000}; Supplementary Table 1). As this dataset does not include many of the mutations found in the full synthetic FoldX dataset, matching subsets of the latter were also generated for comparison.

### Varying synthetic dataset diversity
We explored the importance of dataset diversity, in addition to dataset size, for model performance via the following three metrics:

- The number of antibody clusters, following clustering with a 90% length-matched CDR sequence identity cutoff;
- The number of amino acid substitution types (for example, Arg to Lys);
- The distribution of amino acid substitutions in the complex: mutation locations were classified on the basis of binding partner (antibody or antigen) and proximity to the interface center; for the latter, the interface was divided into two areas (inner and outer shell) defined by concentric circles where, assuming that the interface is approximately flat, the outer

shell circle was defined with a radius $\sqrt{2}$ times the radius of the inner shell circle, to produce two equal areas.

We generated training and validation datasets minimizing and maximizing the different metrics of diversity (Synthetic_FoldX_$\Delta\Delta G$_100000_{sequence/substitution_type/substitution_distribution}_{min/max}; Supplementary Table 1). The test data were kept the same in each case. The respective training, validation and test datasets consisted of 100,000 mutations combined. A 90% length-matched CDR sequence identity cutoff was applied between each.

## Investigating model robustness to noise

We assessed the robustness of our models to noise by (1) shuffling and (2) applying random noise from a Gaussian distribution to the training and validation dataset affinity labels. In each of these cases, the test data remained unmodified:

- Shuffling: Varying percentages of the training and validation $\Delta\Delta G$ dataset labels were shuffled. The effective shuffling percentage was not necessarily equal to the percentage of the dataset that was shuffled, as some labels are the same and others were shuffled back into the same place (Datasets: Synthetic_FoldX_$\Delta\Delta G$_942723_shuffled);
- Gaussian noise: Gaussian noise was applied by adding random values generated from a normal distribution, using numpy.normal, with a set scale (0.5, 1, 2, 5 or 10) to the training and validation datasets (Datasets: Synthetic_FoldX_$\Delta\Delta G$_942723_gaussian_noise).

## Evolutionarily grounded mutations

A recent study demonstrated that the likelihood of FoldX incorrectly predicting a mutation to be stabilizing (in this case, independent of an antigen) could be decreased by up to 11% by limiting FoldX predictions to mutations that are observed naturally[39]. We investigated the effect of limiting our test dataset to such 'evolutionarily grounded' mutations, as defined in ref. 39, on model performance.

We obtained the position-specific scoring matrices (PSSMs) generated from subsets[3] of the Observed Antibody Space database[40,41] and corresponding custom code for calculating log-likelihoods from the authors of ref. 39. As in ref. 39, we defined the evolutionarily grounded mutations as those with a positive log-likelihood and that have a log-likelihood greater than is seen for the WT residue[39].

We mapped the log-likelihood scores to the dataset mutations via the Aho numbering scheme[42] used for the PSSMs, with sequences numbered using ANARCI[36]. There were ten PDBs where ANARCI failed to number a chain with the Aho numbering scheme (3U2S, 4DQO, 4Y5Y, 6BPE, 6E1K, 6OPA, 6U0N, 7EY0, 7LF8 and 7LY9). We applied this approach to mutations from antibody chains from humans or mice, as identified in SAbDab[26,27], as the PSSMs were restricted to these species.

Complexes with human or mouse antibodies made up 75% (710,562 mutations) of the full synthetic dataset. Just over half of these (52%, 366,862) were for mutations to an antibody chain. The final evolutionarily grounded dataset consisted of 47,983 mutations.

For the model performance on the evolutionarily grounded dataset, see the Supplementary Results ('Model performance on evolutionarily grounded mutations').

## Testing Graphinity on modeled structure inputs

We randomly selected 100 mutations (from the Synthetic_FoldX_$\Delta\Delta G$_942723 test dataset, 90% CDR sequence identity cutoff, fold 0 test set) and modeled the structures using Boltz-1[32]. Boltz-1 was run for the WT and mutant complexes, without multiple sequence alignment generation, and excluding mutations that failed due to out-of-memory errors (>300 GB memory).

## Graphinity: EGNN architecture

We developed a deep learning EGNN architecture to predict change in antibody–antigen binding affinity (Fig. 1a). Our model is composed of three $E(n)$ equivariant graph convolutional (EGC) layers[23] with a hidden dimension of 128. The model takes the three-dimensional coordinates of a protein complex structure (PDB file) as input and generates an atomic-resolution graph with nodes representing non-hydrogen atoms and edges representing interactions between nodes <4 Å apart. The node features are a one-hot encoded vector describing the LibMolGrid atom type[43] and the edge features a one-hot encoded vector describing whether the edge is intra- or inter-binding partner. The graphs represent the mutation site neighborhood (for $\Delta\Delta G$ prediction: atoms on the same chain as the mutated residue within 4 Å of the mutated residue (local neighborhood) and atoms on the binding partner chain within 4 Å of these local neighborhood atoms).

For $\Delta\Delta G$ prediction, we generated and aggregated graphs of the WT and mutant structures. Both graphs were fed through the three $E(n)$ EGC layers and the resulting embeddings subtracted from one another (WT − Mutant) before the last linear layer.

The models were trained with mean squared error loss. The architecture was implemented using Python, PyTorch and PyTorch Geometric.

The model parameters were set as follows: optimizer, Adam; learning rate, 0.001; batch size, 32; dropout, 0.2 (applied to edges using torch_geometric.utils.dropout_adj); weight decay, 1e-16; graph readout, global_max_pool over nodes; tanh activation at the output of the coordinate function, True; update coords, True.

Models were trained using PyTorch Lightning for 500 epochs, with the exception of the synthetic FoldX $\Delta\Delta G$ dataset models, which, due to the high computational costs, were trained for 10 epochs. The model checkpoint from the step with the highest validation metric (Pearson's correlation for regression models; ROC AUC for the trastuzumab classification models) was used for testing.

In the models generated with datasets limited to a specific amino acid substitution and transfer learning, we initialized model weights with those from the model trained on the full dataset. In these cases, the learning rate was set to 0.0001.

Model training times are given for training with one graphics processing unit (NVIDIA RTX 6000) and four central processing units on one data fold (80%–10%–10% train–validation–test data split): Experimental_$\Delta\Delta G$_645 (500 epochs), ~1 h; Experimental_$\Delta\Delta G$_608 (500 epochs), ~1 h; Synthetic_FoldX_$\Delta\Delta G$_942723 (10 epochs), ~19.5 h; Synthetic_FoldX_$\Delta\Delta G$_942723 (100 epochs), ~7.25 days; Synthetic_FlexddG_$\Delta\Delta G$_20829 (500 epochs), ~35 h; trastuzumab variants (500 epochs), ~35 h.

## Equiformer

To explore model performance with a different graph-based deep learning architecture, we implemented the Lucidrains Equiformer (Equivariant Graph Attention Transformer)[29,30] for $\Delta\Delta G$ prediction. The model had a hidden dimension of 128, attend_sparse_neighbors = True (allowing for an adjacency matrix input) and otherwise default parameters. The input graphs were generated as for Graphinity (see above), although without edge features included. The graphs were padded to the size of the largest graph in each batch, and a corresponding mask was also provided as input to the model. The WT and mutant graphs were passed through the model, and the resulting embeddings were subtracted from one another (WT − Mutant) before the last linear layer, as was done for Graphinity. The type 0 embedding outputs from the Equiformer model were used in this step. The models were trained with mean squared error loss. The architecture was implemented using Python, PyTorch and PyTorch Geometric.

The model parameters were set as follows: optimizer, Adam; learning rate, 0.001; batch size, 4; dropout, 0.2 (applied to edges using torch_geometric.utils.dropout_adj); weight decay, 1e-16; return_pooled, True; attend_sparse_neighbors, True.

The additional model parameters were left as the default: dim_head, 24; depth, 2; valid_radius, 1e5; reduce_dim_out, False; radial_hidden_dim, 64; attend_self, True; splits, 4; linear_out, True; embedding_grad_frac, 0.5; single_headed_kv, False; ff_include_htype_norms, False; l2_dist_attention, True; reversible, False; gate_attn_head_outputs, True; adj_dim, 0.

The models were trained using PyTorch Lightning for ten epochs. The model checkpoint from the step with the highest validation Pearson's correlation was used for testing.

The model training time is given for training with one graphics processing unit (NVIDIA Ampere A100 80GB) and four central processing units on one data fold (80%–10%–10% train–validation–test data split): Synthetic_FoldX_$\Delta\Delta G$_942723 (10 epochs), ~90 h.

### Structure-informed sequence-based models

Structure-informed sequence-based models were used to explore an alternative model input for the Synthetic_FoldX_$\Delta\Delta G$_942723 dataset. The inputs to these models were 'structure-informed', as they represent the positions at the interfaces of the structures.

The interface residue inputs were ordered on the basis of sequence position for the antibody, followed by the antigen interface positions in the order in which they interact with the antibody positions (linear). The interface residues were provided for the WT antibody, WT antigen, mutant antibody and mutant antigen sequences (in this order and padded to be the same length, respectively). These sequences were then one-hot encoded and used as inputs for ML.

We used these inputs to train FLAML (which considers multiple tree-based model archiectures including XGBoost, LightGBM and random forest models)[28] and a CNN[15]. Default settings were used for the FLAML model and training was allowed for up to 24 h with early stopping enabled. The best model, used for testing, was a LightGBM architecture. The CNN architecture, adapted from the works of Mason et al.[15] and Chinery et al.[44], was trained for ten epochs.

### ESM2 embedding-based model

In addition, a FLAML model was trained on the ESM2[45] embedding of the sequence position that was mutated. The full WT sequences of the antibody and antigen chains of each PDB were extracted and passed through the 3 billion parameter ESM2 model (esm2_t36_3B_UR50D)[45]. Due to the scale of the synthetic mutant dataset, the embeddings were generated for WT sequences only. The 2,560-dimension residue-level embeddings of the WT sequence positions that were mutated were provided as features, along with a one-hot encoding of the chain (antibody heavy chain, antibody light chain or antigen chain), the numerical sequence position and the one-hot encoding of the WT and mutant amino acids. Default settings were used to train a FLAML model on these data, and training was allowed for up to 24 h with early stopping enabled. The best model, used for testing, was a LightGBM architecture.

### Tree-based model trained on featurized structures

We generated a tree-based model trained on featurized structures. Features were derived from the antibody–antigen structures as in mCSM-AB2[18], described below. For each feature, we calculated the difference between the values for the WT and mutant structures (WT − Mutant):

- FoldX AnalyseComplex energetic terms: We used the FoldX AnalyseComplex function[11] to calculate interaction energetic terms for the WT and mutant complexes.
- Arpeggio interactions: We calculated the inter-protein interface interactions (for example, H-bonds and ionic interactions) of the complexes using Arpeggio[46].
- Pharmacophore vectors: To represent the change in amino acid upon mutation, we calculated a change in the pharmacophore counts, adapted from ref. [47]. We assigned pharmacoph-

ores (for example, hydrophobic, H-bond acceptor or H-bond donor) to each atom in each amino acid and took a sum across the amino acid (Supplementary Table 5). To note, an atom can have more than one pharmacophore.
- Buried surface area: We calculated the buried surface area (BSA) for each binding partner (antibody and antigen) in each complex using the PSA program[48]: BSA = $SA_{free}$ − $SA_{bound}$. An average change in BSA across the two binding partners was calculated.
- PSSM evolutionary term: A measure of residue conservation at a position was captured in PSSMs. We calculated the PSSM scores using PSI-BLAST[49] (parameters: evolutionary scoring matrix = PAM30, num_iterations = 3, evalue = 1E-10, seg = Yes, comp_based_stats = 1, and db = swissprot) as in ref. [18].

We generated ExtraTrees regression models using the Python scikit-learn ExtraTreesRegressor package with 300 estimators (as in ref. [18]) and remaining default parameters. This is not a direct comparison with the mCSM-AB2 model, as we do not incorporate the graph-based features of the CSM-based models.

We applied this featurization and subsequent ExtraTrees model to the Experimental_$\Delta\Delta G$_608 dataset. Given the time required for featurization, it was computationally infeasible to apply this approach to the large synthetic dataset.

### Comparison against protein–protein interaction $\Delta\Delta G$ prediction methods

We retrained DGCddG[24] and RDE-PPI Network[25] on the Experimental_$\Delta\Delta G$_645 and Experimental_$\Delta\Delta G$_608 datasets (90% CDR sequence identity cutoff between train, validation and test datasets). We installed these methods from GitHub (https://github.com/lennylv/DGCddG and https://github.com/luost26/RDE-PPI, respectively) and used training scripts from the repositories (prot-cv/cv_fold_645.py and train_rde_network_skempi.py, respectively) with default parameters. For RDE-PPI, we started training from the pretrained model provided by the authors (RDE.pt). In addition, we updated the dataloader (rde/dataset/skempi.py) to be compatible with nonrandomly split dataset folds. For both methods, the model checkpoint with the lowest validation loss was used for testing.

DGCddG requires featurized inputs generated from BLAST and HHblits profiles. We generated these profiles using PSI-BLAST[49] (version 2.16.0) with the Swiss-Prot database[50] (26 February 2025; as recommended by the authors in personal communication) and HHblits (HH-suite3[51]).

We also trained the RDE-PPI Network method on the synthetic dataset (Synthetic_FoldX_$\Delta\Delta G$_942723), using the approach described above. Implementing DGCddG for a dataset of this size was computationally infeasible, however, owing to the time required to generate the BLAST and HHblits profiles.

### Trastuzumab variants

We obtained the dataset of trastuzumab CDRH3 variants and corresponding binary binding labels from Mason et al.[15]. The sequences were mutated at ten amino acid positions in the CDRH3[15]. The variants that had been labeled as both binding and non-binding were assigned the binding label, as in ref. [15]. This resulted in 36,391 variants, 11,277 of which were labeled as binding. We split the dataset (1) randomly using sklearn.model_selection.train_test_split and (2) with a clonotype plus sequence-identity split. For (2), variants were clustered on the basis of the V- and J-gene assignments, as labeled by ANARCI[36], and sequence identity of the CDRH3 (limited to the ten mutated positions). Sequence identity in this case describes the maximum allowed edit distance from a representative sequence (cluster center). For example, a minimum identity of 70% allows edit distances of up to three residues from the cluster center. We used the clonotype and sequence identity approach

as CD-HIT did not run with the ten-position variant sequences owing to their short length.

The trastuzumab datasets were prepared with a 70%–15%–15% train–validation–test split to allow comparison with ref. 15.

We modeled structures of the trastuzumab variants in complex with HER2 using the FoldX BuildModel function starting from a FoldX-'repaired' structure of PDB 1N8Z[11,52]. Although this approach is unlikely to capture the true structural effect of the mutations, as FoldX does not model changes to the backbone[53], it is fast and allows us to avoid docking by starting from a structure of a bound complex.

We adapted the Graphinity architecture for this task. The input was changed to be one graph only (and subsequently there was no subtraction of embeddings before the final layer), and the graphs were formed from the ten mutated CDRH3 residues and surrounding neighborhood (antibody atoms within 4 Å of CDRH3 atoms (antibody neighborhood), antigen atoms within 4 Å of the antibody neighborhood and antigen atoms within 4 Å of these antigen atoms). We also updated the model for classification by changing the loss function (to binary cross-entropy with logits) and accuracy metrics (to ROC AUC and AP).

## Statistics and reproducibility
The Pearson's and Spearmans' rank correlations were calculated using scipy.stats.pearsonr and scipy.stats.spearmanr, respectively.

No statistical method was used to predetermine sample size. No data were excluded from the analyses. The experiments were not randomized. The investigators were not blinded to allocation during experiments and outcome assessment.

## Data visualization
Figures were generated using Python, matplotlib, seaborn, PyMOL and PowerPoint. Colors were selected in part using ColorBrewer 2.0 (https://colorbrewer2.org).

## Reporting summary
Further information on research design is available in the Nature Portfolio Reporting Summary linked to this article.

## Data availability
The synthetic $\Delta\Delta G$ datasets are available via GitHub at https://github.com/oxpig/Graphinity and via Zenodo at https://doi.org/10.5281/zenodo.15384945 (ref. 54). The corresponding PDB files are available via Zenodo at https://doi.org/10.5281/zenodo.15384945 (ref. 54) (excluding PDB files for the FoldX mutant complexes, owing to dataset size restrictions) and at https://opig.stats.ox.ac.uk/data/downloads/affinity_dataset/ (all PDB files). The AB-Bind database[20] is available via GitHub at https://github.com/sarahsirin/AB-Bind-Database. The SKEMPI v2.0 database[31] is available at https://life.bsc.es/pid/skempi2. Source data are provided with this paper. These data are also available via GitHub at https://github.com/oxpig/Graphinity.

## Code availability
The Graphinity EGNN model code is available via GitHub at https://github.com/oxpig/Graphinity and via Zenodo at https://doi.org/10.5281/zenodo.15237531 (ref. 55).

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

## Acknowledgements

This work was supported by the UKRI Medical Research Council (grant number MR/N013468/1 awarded to A.M.H.), the UKRI Engineering and Physical Sciences Research Council (grant number EP/L016044/1 awarded to C.S.), the UKRI Biotechnology and Biological Sciences Research Council (grant number BB/V509681/1 awarded to L.C.), AstraZeneca and GlaxoSmithKline. We thank C. Outeiral, who compiled the Rosetta version that was used to generate the Flex ddG synthetic dataset.

## Author contributions

A.M.H. and C.S. developed the code. A.M.H. and C.M.D. designed the experiments. A.M.H. performed the experiments and analyses. L.C. prepared the trastuzumab data. A.M.H. wrote the paper, and all other authors revised it. C.M.D. supervised the work. All authors read and approved the final version of the paper.

## Competing interests

C.M.D. discloses membership of the Scientific Advisory Board of Fusion Antibodies and AI Proteins, as well as a founder of Dalton. The other authors declare no competing interests.

## Additional information

**Correspondence and requests for materials** should be addressed to Alissa M. Hummer or Charlotte M. Deane.

reports are available. Primary Handling Editor: Kaitlin McCardle, in collaboration with the *Nature Computational Science* team.

# Reporting Summary

## Statistics

For all statistical analyses, confirm that the following items are present in the figure legend, table legend, main text, or Methods section.

| n/a | Confirmed | |
|---|---|---|
| ☐ | ☒ | The exact sample size (*n*) for each experimental group/condition, given as a discrete number and unit of measurement |
| ☒ | ☐ | A statement on whether measurements were taken from distinct samples or whether the same sample was measured repeatedly |
| ☒ | ☐ | The statistical test(s) used AND whether they are one- or two-sided *Only common tests should be described solely by name; describe more complex techniques in the Methods section.* |
| ☒ | ☐ | A description of all covariates tested |
| ☒ | ☐ | A description of any assumptions or corrections, such as tests of normality and adjustment for multiple comparisons |
| ☒ | ☐ | A full description of the statistical parameters including central tendency (e.g. means) or other basic estimates (e.g. regression coefficient) AND variation (e.g. standard deviation) or associated estimates of uncertainty (e.g. confidence intervals) |
| ☒ | ☐ | For null hypothesis testing, the test statistic (e.g. *F*, *t*, *r*) with confidence intervals, effect sizes, degrees of freedom and *P* value noted *Give P values as exact values whenever suitable.* |
| ☒ | ☐ | For Bayesian analysis, information on the choice of priors and Markov chain Monte Carlo settings |
| ☒ | ☐ | For hierarchical and complex designs, identification of the appropriate level for tests and full reporting of outcomes |
| ☐ | ☒ | Estimates of effect sizes (e.g. Cohen's *d*, Pearson's *r*), indicating how they were calculated |

*Our web collection on statistics for biologists contains articles on many of the points above.*

## Software and code

Policy information about availability of computer code

| Data collection | We modeled mutations and predicted the corresponding change in binding affinity values for antibody-antigen complexes from the SAbDab database (accession date 19 May 2022) using FoldX (version 5) and Rosetta Flex ddG (Rosetta version 2020.08+release.cb1caba). |
|---|---|
| Data analysis | We built the EGNN deep learning models in Python v3.7.10 using PyTorch v1.8.0 and PyTorch Geometric v1.6.3. The code is available at https://github.com/oxpig/Graphinity. We visualized the data using Python v3.9.1, matplotlib v3.9.2, seaborn v0.13.2, PyMOL v2.3.0, and PowerPoint v16.95.4. Colors were selected in part using ColorBrewer 2.0 (https://colorbrewer2.org). |

For manuscripts utilizing custom algorithms or software that are central to the research but not yet described in published literature, software must be made available to editors and reviewers. We strongly encourage code deposition in a community repository (e.g. GitHub). See the Nature Portfolio guidelines for submitting code & software for further information.

## Data

Policy information about availability of data

All manuscripts must include a data availability statement. This statement should provide the following information, where applicable:

- Accession codes, unique identifiers, or web links for publicly available datasets
- A description of any restrictions on data availability
- For clinical datasets or third party data, please ensure that the statement adheres to our policy

The synthetic ΔΔG datasets and links to download the corresponding PDBs can be found at https://github.com/oxpig/Graphinity.

The AB-Bind database is available at https://github.com/sarahsirin/AB-Bind-Database. The SKEMPI v2.0 database is available at https://life.bsc.es/pid/skempi2.

Source data for Figures 2-4 is available with this manuscript and at https://github.com/oxpig/Graphinity.

## Human research participants

Policy information about studies involving human research participants and Sex and Gender in Research.

| | |
|---|---|
| Reporting on sex and gender | NA – no studies involving human participants were conducted for this work. |
| Population characteristics | NA – no studies involving human participants were conducted for this work. |
| Recruitment | NA – no studies involving human participants were conducted for this work. |
| Ethics oversight | NA – no studies involving human participants were conducted for this work. |

Note that full information on the approval of the study protocol must also be provided in the manuscript.

# Field-specific reporting

Please select the one below that is the best fit for your research. If you are not sure, read the appropriate sections before making your selection.

☒ Life sciences ☐ Behavioural & social sciences ☐ Ecological, evolutionary & environmental sciences

For a reference copy of the document with all sections, see nature.com/documents/nr-reporting-summary-flat.pdf

# Life sciences study design

All studies must disclose on these points even when the disclosure is negative.

| | |
|---|---|
| Sample size | We created synthetic change in binding affinity datasets using PDBs from SAbDab. The final datasets consisted of 942,723 FoldX mutations from 1471 antibody-antigen complexes and 20,829 Flex ddG mutations from 1302 antibody-antigen complexes.<br><br>The synthetic FoldX dataset size was determined as the number of mutations that could be generated through exhaustive mutagenesis of interface positions from a clustered dataset (90% length-matched CDR sequence identity cutoff) of solved antibody-antigen complex structures. The synthetic Flex ddG dataset was subsampled from the synthetic FoldX dataset at ca. 16 mutations per antibody-antigen complex. The Flex ddG dataset size was restricted by computational requirements.<br><br>The sufficiency of these datasets was analyzed through developing models on subsampled datasets. Performance plateaued as the full dataset size was reached.<br><br>Analyses were also conducted on existing experimental databases, AB-Bind and SKEMPI 2.0. |
| Data exclusions | We clustered the SAbDab PDBs based on 90% length-matched CDR sequence identity and carried forward one PDB from each cluster for interface mutagenesis. We excluded mutations where the WT amino acid was 'X', the chain identifier was a number, the antibody and antigen were > 4 A apart and the FoldX Interaction Energy calculation or Rosetta Flex ddG mutation modeling failed. |
| Replication | The FoldX software is deterministic. |
| Randomization | This study is centered on the development and validation of models for predicting change in antibody-antigen binding affinity. The quantitative data is formatted as the difference in binding affinity between the wild-type and mutant complexes (ΔΔG) and does not test for a causal relationship between groups. As such, randomization was not applicable to this study. |
| Blinding | This study is centered on the development and validation of models for predicting change in antibody-antigen binding affinity. The quantitative data is formatted as the difference in binding affinity between the wild-type and mutant complexes (ΔΔG) and does not test for a |

causal relationship between groups. As such, blinding was not applicable to this study.

# Reporting for specific materials, systems and methods

We require information from authors about some types of materials, experimental systems and methods used in many studies. Here, indicate whether each material, system or method listed is relevant to your study. If you are not sure if a list item applies to your research, read the appropriate section before selecting a response.

## Materials & experimental systems

| n/a | Involved in the study |
|---|---|
| ☒ ☐ | Antibodies |
| ☒ ☐ | Eukaryotic cell lines |
| ☒ ☐ | Palaeontology and archaeology |
| ☒ ☐ | Animals and other organisms |
| ☒ ☐ | Clinical data |
| ☒ ☐ | Dual use research of concern |

## Methods

| n/a | Involved in the study |
|---|---|
| ☒ ☐ | ChIP-seq |
| ☒ ☐ | Flow cytometry |
| ☒ ☐ | MRI-based neuroimaging |

