## [Peer Review File · Nature Computational Science]

Investigating the Volume and Diversity of Data Needed for Generalizable Antibody-Antigen $\Delta\Delta G$ Prediction

Corresponding Author: Dr Alissa Hummer

Version 0:

Decision Letter:

**** Please ensure you delete the link to your author homepage in this e-mail if you wish to forward it to your co-authors. ****

Dear Dr Hummer,

Your manuscript "Investigating the Volume and Diversity of Data Needed for Generalizable Antibody-Antigen $\Delta\Delta G$ Prediction" has now been seen by 3 referees, whose comments are appended below. You will see that while they find your work of interest, they have raised points that need to be addressed before we can make a decision on publication.

The referees' reports seem to be quite clear. Naturally, we will need you to address ***all*** of the points raised.

While we ask you to address all of the points raised, the following points need to be substantially worked on:

- 1) Please be sure to provide comparisons to existing methods in the field.
- 2) Please be sure to provide all of the details regarding your model, training and testing.

Please use the following link to submit your revised manuscript and a point-by-point response to the referees' comments (which should be in a separate document to any cover letter):

Link Redacted

**** This url links to your confidential homepage and associated information about manuscripts you may have submitted or be reviewing for us. If you wish to forward this e-mail to co-authors, please delete this link to your homepage first. ****

To aid in the review process, we would appreciate it if you could also provide a copy of your manuscript files that indicates your revisions by making use of Track Changes or similar mark-up tools. Please also ensure that all correspondence is marked with your Nature Computational Science reference number in the subject line.

In addition, please make sure to upload a Word Document or LaTeX version of your text, to assist us in the editorial stage.

To improve transparency in authorship, we request that all authors identified as 'corresponding author' on published papers create and link their Open Researcher and Contributor Identifier (ORCID) with their account on the Manuscript Tracking System (MTS), prior to acceptance. ORCID helps the scientific community achieve unambiguous attribution of all scholarly contributions. You can create and link your ORCID from the home page of the MTS by clicking on 'Modify my Springer Nature account'. For more information please visit www.springernature.com/orcid.

We hope to receive your revised paper within three weeks. If you cannot send it within this time, please let us know.

Best regards,

Reviewers comments:

Reviewer #1 (Remarks to the Author):

Hummer et al constructed a deep learning model, called Graphinity for predicting the ddGs of protein-protein interactions (PPIs). The ddGs of PPIs have been highly desired because PPIs are critical for rational protein engineering (for drug development) as well as fundamental biological functions of proteins. The authors found Graphinity became overtrained with the hundreds of data points currently available. They then generated synthetic datasets using FoldX and Rosetta and trained the model using the synthetic datasets. In this trial, the authors found that a larger (~ with orders of magnitude more) and more diverse PPI dataset is needed to construct a general and robust prediction model of PPIs.

Overall, the conclusion is well supported by the results. The reviewer personally thinks that a current bottleneck for AI model construction is the size and diversity of a training dataset, and the authors successfully proved this idea and showed the size and diversity bottlenecks in a quantitative manner.

However, this is also the weakest point of this study: they constructed a prediction model of 'predicted' ddG of PPI. Historically, the limitation of datasets has been a huge problem in this field and researchers have been trying to solve this issue. In this term, this study failed to solve the issue (or even propose a hint to do so) but just specified it, which limits the range of potential readers. (They did not compare the accuracy of Graphinity with the many previous PPI prediction models, either.) Therefore, the reviewer thinks this should be published in a more specific journal.

Reviewer #2 (Remarks to the Author):

Hummer, Deane, and colleagues present a Hummer, Deane, and colleagues develop an equivariant graph neural network (called 'Graphinity') to predict DDG values for antibody-antigen complexes. The prediction of binding affinity upon antibody sequence change (DDG) is perhaps the biggest open challenge for antibody recognition; there are experimental papers which provide solutions for individual antibodies using massive experimental data, the current state of the art yields overall poor performance for antibodies outside of their immediate training set. The major value - and impact- of this paper is by supplying a lower bound on the amount and diversity of the experimentally data required. This is of considerable importance to the broader field, as it gives the blueprint for machine learning ready data to be generated. The authors first show that the existing highly validated, openly available experimental datasets lead to model overtraining. They then generate synthetic data using the fixed backbone prediction program FoldX and the limited backbone movement Rosetta app 'FlexDDG'. For the FoldX dataset, they sample ~700 mutations each for 1471 antibody-antigen complexes to generate a dataset. They find that their model can recapitulate FoldX performance, while their model shows a comparatively worse performance for FlexDDG, even when controlling for the size of the training set. I have the following small comments.

1. For FoldX - the DDG scores are of a different magnitude than the distribution shown in S Fig 1c. Specifically, in Figure 2c, and in Figure 4. The distribution shows a kernel density tailoff between -4 and -6 kcal/mol, while the main text figures show data points out to nearly -100 kcal/mol. Perhaps the discrepancy can be rationalized by the sheer amount of simulated data, such that small numbers of outliers become apparent? While inclusion of this (small amount of) data should not impact the regression values reported, the results at the tails of these distributions are physically unrealistic. Ideally, these values should be truncated with a softmax or other function before comparing regressions. Alternatively, since this is just synthetic data, a brief description in the text on what are likely non-binding sequences would guide the reader.

2. For S. Fig 9, it appears that the model over-performed on mutations to aromatics and underperforms on smaller, polar residues. This suggests that experimental sampling should be biased to mutations to smaller residues. This is important because later in affinity maturation campaigns for antibodies, smaller polar residues often pop up. Perhaps the authors could comment on this finding, or explain why it is not relevant.

3. As a quick technical note - (not requiring author response), there are at least two hypotheses for why Graphinity underperforms on FlexDDG compared with the FoldX datasets, controlling for size of the training set. First, FlexDDG moves the backbone (albeit modestly), and these changes may be harder to capture. Second, for FlexDDG the point of nstruct=35 (rather than nstruct=1) is that the measurement itself is pretty noisy and needs multiple structures to average. Since generating 35 decoys per mutation is prohibitive, there aren't great rigorous ways that I can think of to quantify this noise.

4. Supporting Methods. In "Model training times" - should include details for the the Synthetic_FlexDDG dataset.

Reviewer #2 (Remarks on code availability):

I checked to ensure that the Github was accessible and available, and also looked at the README for Graphinity which contains detailed instructions for installing the application. I also looked at the folders containing the synthetically generated

data.

Reviewer #3 (Remarks to the Author):

Predicting the change in the binding free energy DDG of antibody-antigen complexes upon mutation is critical in many biological and medical problems. However, related machine learning techniques suffer from the poor availability of experimental data needed for their training and testing. The manuscript introduces an equivariant graph neural network, called Graphinity, to predict DDG from protein structures and tests it on synthetic complexes. The result is that the amount of available data is too small to achieve good performance with this and all competing algorithms; however, Graphinity will potentially outperform the others when a sufficient amount of data is available.

In my opinion, the paper is good and of interest to the community working not only in machine learning but also in antibody design. Therefore, it might deserve publication, although I do have some criticisms, mainly related to the explanation of the results.

1) In general, I interpret the training and testing of the algorithm on synthetic data generated by FoldX and Rosetta as a sandbox to explore its potential, on the assumption that the data has similar characteristics to real data. For this strategy to work, the data does not have to be indistinguishable from real data, just plausible. Indeed, the DDG predicted by FoldX and Rosetta correlate poorly with experimental values. Nevertheless, these two models are based on a physical description of the interactions that stabilise proteins and, although the parameterisation is far from perfect, they recapitulate the main physical properties of the system. I think this point is not clear from the start and should be emphasised more in the introduction to the manuscript. If one misses this point, one is likely to conclude that the model is not predictive of experimental DDG, which (if I understand correctly) is not the goal of this work. The comment of the degree of realism of FoldX (line 157) does not help in this respect. Also (line 186), the further use of Rosetta should not be associated with a more or less realistic dataset, but simply with a different parameterisation, to show that the algorithm is robust if a plausible but different dataset is used.

2) I do not understand the decision to assign a DDG=-8 kcal/mol to non-binding mutations. Why this? How do the results depend on this choice?

3) After the thorough analysis in the manuscript, I cannot find a clear statement in the discussion about the minimum size of the experimental dataset needed to get reasonable results in DDG prediction.

4) An interesting point, if easily achievable, would be whether using predicted structures instead of crystallographic ones significantly changes the performance of the predictor, at least within the "sandbox".

Version 1:

Decision Letter:

Our ref: NATCOMPUTSCI-24-2831A

31st March 2025

Dear Dr. Hummer,

Thank you for submitting your revised manuscript "Investigating the Volume and Diversity of Data Needed for Generalizable Antibody-Antigen $\Delta\Delta G$ Prediction" (NATCOMPUTSCI-24-2831A). It has now been seen by the original referees and their comments are below. The reviewers find that the paper has improved in revision, and therefore we'll be happy in principle to publish it in Nature Computational Science, pending minor revisions to satisfy the referees' final requests and to comply with our editorial and formatting guidelines.

TRANSPARENT PEER REVIEW

Nature Computational Science offers a transparent peer review option for original research manuscripts. We encourage increased transparency in peer review by publishing the reviewer comments, author rebuttal letters and editorial decision letters if the authors agree. Such peer review material is made available as a supplementary peer review file. **Please remember to choose, using the manuscript system, whether or not you want to participate in transparent peer review.**

Please note: we allow redactions to authors' rebuttal and reviewer comments in the interest of confidentiality. If you are concerned about the release of confidential data, please let us know specifically what information you would like to have removed. Please note that we cannot incorporate redactions for any other reasons. Reviewer names will be published in the peer review files if the reviewer signed the comments to authors, or if reviewers explicitly agree to release their name. For more information, please refer to our <https://www.nature.com/documents/nr-transparent-peer-review.pdf> target="new">FAQ page.

Thank you again for your interest in Nature Computational Science. Please do not hesitate to contact me if you have any questions.

Sincerely,

Kaitlin McCardle, PhD
Senior Editor
Nature Computational Science

ORCID

Reviewer #1 (Remarks to the Author):

The reviewer recognized the substantial efforts for the revision. According to Table 1 and S3, Graphinity is almost comparable to the previous PPI prediction model or a bit worse than the existing models. If the reviewer understands the performance correctly, can the author say "Graphinity, an equivariant graph neural network architecture built directly from antibody-antigen structures that achieves state-of-the-art performance on experimental $\Delta\Delta G$ prediction."? If the authors think "our work to provide a lower bound on the amount of data needed, which would frame future method development and data collection efforts", it would be better to emphasize this point in the abstract as well.

Reviewer #2 (Remarks to the Author):

The rebuttal and revised manuscript satisfies my minor issues raised.

Reviewer #3 (Remarks to the Author):

The authors addressed all my previous comments. I think the manuscript may be accepted for publication.

Version 2:

Decision Letter:

Dear Dr Hummer,

We are pleased to inform you that your Article "Investigating the Volume and Diversity of Data Needed for Generalizable Antibody-Antigen $\Delta\Delta G$ Prediction" has now been accepted for publication in Nature Computational Science.

Once your manuscript is typeset, you will receive an email with a link to choose the appropriate publishing options for your paper and our Author Services team will be in touch regarding any additional information that may be required.

Acceptance of your manuscript is conditional on all authors' agreement with our publication policies (see <https://www.nature.com/natcomputsci/for-authors>). In particular your manuscript must not be published elsewhere and there must be no announcement of the work to any media outlet until the publication date (the day on which it is uploaded onto our web site).

Before your manuscript is typeset, we will edit the text to ensure it is intelligible to our wide readership and conforms to house style. We look particularly carefully at the titles of all papers to ensure that they are relatively brief and understandable.

Once your manuscript is typeset, you will receive a link to your electronic proof via email with a request to make any corrections within 48 hours. If, when you receive your proof, you cannot meet this deadline, please inform us at rjsproduction@springernature.com immediately.

If you have queries at any point during the production process then please contact the production team at rjsproduction@springernature.com.

We welcome the submission of potential cover material (including a short caption of around 40 words) related to your manuscript; suggestions should be sent to Nature Computational Science as electronic files (the image should be 300 dpi at 210 x 297 mm in either TIFF or JPEG format). We also welcome suggestions for the Hero Image, which appears at the top of our [home page](http://www.nature.com/natcomputsci); these should be 72 dpi at 1400 x 400 pixels in JPEG format. Please note that such pictures should be selected more for their aesthetic appeal than for their scientific content, and that colour images work better than black and white or grayscale images. Please do not try to design a cover with the Nature Computational Science logo etc., and please do not submit composites of images related to your work. I am sure you will understand that we cannot make any promise as to whether any of your suggestions might be selected for the cover of the journal.

Best regards,

Kaitlin McCardle, PhD
Senior Editor
Nature Computational Science

P.S. Click on the following link if you would like to recommend Nature Computational Science to your librarian: <https://www.springernature.com/gp/librarians/recommend-to-your-library>

** Visit the Springer Nature Editorial and Publishing website at <http://editorial-jobs.springernature.com> for more information about our career opportunities. If you have any questions please click [here](mailto:editorial.publishing.jobs@springernature.com). **

Your manuscript "Investigating the Volume and Diversity of Data Needed for Generalizable Antibody-Antigen $\Delta\Delta G$ Prediction" has now been seen by 3 referees, whose comments are appended below. You will see that while they find your work of interest, they have raised points that need to be addressed before we can make a decision on publication.

The referees' reports seem to be quite clear. Naturally, we will need you to address **all of the points raised.**

While we ask you to address all of the points raised, the following points need to be substantially worked on:

1) Please be sure to provide comparisons to existing methods in the field.

2) Please be sure to provide all of the details regarding your model, training and testing.

We thank the editor for the helpful comments and consideration of our manuscript. We have addressed these points in more detail below, in particular, point 1) in the response to Reviewer 1 and point 2) in the response to Reviewer 2.

Reviewers comments:

Reviewer #1 (Remarks to the Author):

Hummer et al constructed a deep learning model, called Graphinity for predicting the ddGs of protein-protein interactions (PPIs). The ddGs of PPIs have been highly desired because PPIs are critical for rational protein engineering (for drug development) as well as fundamental biological functions of proteins. The authors found Graphinity became overtrained with the hundreds of data points currently available. They then generated synthetic datasets using FoldX and Rosetta and trained the model using the synthetic datasets. In this trial, the authors found that a larger (~ with orders of magnitude more) and more diverse PPI dataset is needed to construct a general and robust prediction model of PPIs.

Overall, the conclusion is well supported by the results. The reviewer personally thinks that a current bottleneck for AI model construction is the size and diversity of a training dataset, and the authors successfully proved this idea and showed the size and diversity bottlenecks in a quantitative manner.

We thank the reviewer for the supportive comments and for highlighting the importance of dataset construction for machine learning model development.

However, this is also the weakest point of this study: they constructed a prediction model of 'predicted' ddG of PPI. Historically, the limitation of datasets has been a huge problem in this field and researchers have been trying to solve this issue. In this term, this study failed to solve the issue (or even propose a hint to do so) but just specified it, which limits the range of potential readers.

We thank the reviewer for this suggestion and agree that it would be helpful to discuss a strategy for solving this data problem. We intended for our work to provide a lower bound on the amount of data needed, which would frame future method development and data collection efforts. Obtaining a dataset of this size, however, is a significant challenge that will require a community-wide effort and many components. For example, we envision that higher throughput of experimental methods, a collaborative approach to dataset design, the creation of a standardized data repository, and robust evaluation of ML models will be needed. As such, we originally felt this was beyond the scope of this manuscript.

We have now updated the Discussion to include some broader suggestions. Additionally, we have included more detail on our thoughts for potential strategies below and would be eager to engage with the reviewer and wider community on approaches to implement these.

- 1) Experimental method development to increase throughput for antibody-antigen affinity quantification. These methods must be highly scalable and produce measurements that are comparable between different antibodies and experiments (ideally K_D values, rather than proxies). There have been advances in this area in recent years; for example, the MAGMA-seq method could be applied to tens of thousands of antibody variants (Petersen et al., 2024; <https://doi.org/10.1038/s41467-024-48072-z>).
- 2) Dataset design. To ensure the generation of a robust dataset, it would be helpful to have a collective approach for dataset design (rather than scraping together different $\Delta\Delta G$

measurements from across literature, which is the current status of the field). The planned dataset should be compared against diversity metrics of antibody sequences (e.g., aiming to cover a certain percentage of the sequence diversity in the PDB or Observed Antibody Space and capture all amino acid mutation types). Further considerations include data subsets that are more challenging for existing ML models (e.g., particular areas of sequence/structure space, mutations to small amino acids). Additionally, dataset planning could be enhanced through community input on therapeutically/biotechnologically relevant antibody and antigen sequences.

- 3) Data repository to store $\Delta\Delta G$ values in a standardized format with metadata. Collections of $\Delta\Delta G$ values from literature, such as AB-Bind and SKEMPI, have formed the basis of nearly all ML development in this field. Expanding beyond these datasets, which were time-consuming to collate, there is a need for a unified database that is regularly updated and maintained. It would be beneficial to encourage or require publications with $\Delta\Delta G$ measurements to deposit their values here. Beyond $\Delta\Delta G$ values, metadata should also be recorded. Affinity measurements are impacted by experimental conditions (e.g., temperature, pH, ionic strength, buffer composition) and, ideally, future ML methods will be able to leverage this information for improved predictions.
- 4) Robust ML model evaluation. A blind assessment would be the optimal approach to achieve robust and unbiased evaluation of ML methods. The AAntibody challenge (Erasmus et al., 2024; <https://doi.org/10.1038/s41587-024-02469-9>), which included categories for *in silico* affinity maturation and affinity rank prediction, lays a foundation for this type of assessment. Additionally, model evaluation could be organized in a manner akin to, or even part of, CASP (Critical Assessment of Structure Prediction). Antibody-antigen complex structure prediction and protein–small molecule ligand binding affinity prediction were included in the most recent CASP competition, indicating this could be another successful avenue for antibody-antigen affinity prediction evaluation.

Updates to the text (“Discussion”):

Our results highlight the need to move towards ‘machine learning-grade data’, where model development **informs** the data generation process. **Additionally, several key efforts should be considered to advance generalizable affinity prediction: (1) increasing the throughput of experimental methods for affinity measurement (e.g., [41]), (2) designing diverse and well-structured datasets, (3) establishing a standardized, regularly updated repository for $\Delta\Delta G$ values and metadata, and (4) implementing robust, blind assessments of model performance (e.g., [42]). These measures would ultimately improve the reliability of machine learning for antibody-antigen affinity prediction.**

(They did not compare the accuracy of Graphinity with the many previous PPI prediction models, either.) Therefore, the reviewer thinks this should be published in a more specific journal.

We appreciate the reviewer’s comment and the importance of comparing against existing methods.

We had originally been unable to implement a head-to-head comparison against antibody-focused $\Delta\Delta G$ prediction methods that had been specifically benchmarked on the AB-Bind dataset. The mCSM-based methods (Pires et al., 2016; Myung et al., 2020) are only available as web

servers. The TopNetTree model (Wang et al., 2020) relies on an outdated software package (MIBPB), which could not be installed due to dependency issues. As a result, neither method could be tested or retrained with a more stringent train-test split. The supplementary results for TopNetTree, however, clearly indicate this model suffers from overtraining: a leave-one-complex-out cross-validation test produces an average Pearson's correlation of only 0.17 (compared to 0.65 with a random split). Additionally, both of these methods require time-consuming featurization that limits the scale of datasets they can be applied to.

We have now expanded the scope to compare against general protein predictors of $\Delta\Delta G$. We identified the methods listed below. If the reviewer knows of further methods that offer distinct approaches or significant improvements, we would be happy to consider these as well.

- DGCddG (Jiang et al., 2023; <https://doi.org/10.1109/tcbb.2022.3233627>)
- RDE-PPI (Luo et al., 2024; <https://doi.org/10.1101/2023.02.28.530137>)
- DiffAffinity (Liu et al., 2023; <https://openreview.net/forum?id=BGP5Vjt93A>)
- Prompt-DDG (Wu et al., <https://doi.org/10.48550/arXiv.2405.10348>)
- Light-DDG (Wu et al., <https://doi.org/10.48550/arXiv.2502.06913>)
- DDAffinity (Yu et al., 2024, <https://doi.org/10.1093/bioinformatics/btae232>)
- GeoPPI (Liu et al., <https://doi.org/10.1371/journal.pcbi.1009284>)
- DDGPred (<https://github.com/HeliXonProtein/binding-ddg-predictor>)
- PIANO (Zhang et al., 2024; <https://doi.org/10.1038/s42003-024-07066-9>)
- DDMut-PPI (Zhou et al., 2024; <https://doi.org/10.1093/nar/gkae412>)
- Transfer-DDG (Wang et al., 2023; <https://doi.org/10.1109/ONCON60463.2023.10430682>)
- DDGemb (Savojarado et al., 2025; <https://doi.org/10.1093/bioinformatics/btaf019>)

Given the challenges posed by overtraining, however, we focused our comparisons on methods for which training code had been made available. We wanted to retrain the models with effective train-validation-test dataset cutoffs to ensure robust and fair evaluation. For some methods, no code (DDMut-PPI, Transfer-DDG, DDGemb) or no training code (GeoPPI, DDGPred, PIANO) was provided.

After downloading and setting up the repositories for the remaining methods, we found further issues for DiffAffinity, Prompt-DDG, Light-DDG, and DDAffinity that restricted us from retraining.

- DiffAffinity (environment issue): DiffAffinity relies on another repository, *riemannian-score-sde* (<https://github.com/oxcsm/riemannian-score-sde>), which could not be installed. Multiple issues have been raised on the GitHub repositories for DiffAffinity and *riemannian-score-sde* (e.g., <https://github.com/EureKaZhu/DiffAffinity/issues/4>), including by the author of DiffAffinity (e.g., <https://github.com/oxcsm/riemannian-score-sde/issues/5>). No updates have been made to the repositories to address these installation difficulties, however.
- Prompt-DDG and Light-DDG (missing pre-training models and data): These two methods were developed through pre-training (on 'microenvironmental patterns' and synthetic data

from Prompt-DDG, respectively) followed by training for $\Delta\Delta G$ prediction. The authors, however, did not make the pre-trained models available. The authors of Prompt-DDG commented on this in a GitHub issue (“We only provided pre-training [*sic; training*] weights for the DDG_model and not for the VQ-VAE.”; <https://github.com/LirongWu/Prompt-DDG/issues/3>). Additionally, they indicated their code cannot be readily applied to new complexes: “If you want to use Prompt-DDG for prediction of any of the [*sic; other*] protein complexes, you need to write a dataloader yourself ..., which requires quite a bit of work.”; <https://github.com/LirongWu/Prompt-DDG/issues/4>).

- DDAffinity (missing data processing code): While the authors made part of the training and testing code available, they did not provide the data preprocessing code to allow the model to be applied to new structures. In a recent GitHub issue raising this question, the authors indicated they plan to upload this code but have not yet done so (<https://github.com/ak422/DDAffinity/issues/7>).

We were able to successfully implement and retrain the DGCddG and RDE-PPI (Network) methods on the experimental datasets (10-fold cross-validation with a 90% CDR sequence identity train-validation-test cutoff). RDE-PPI involves a pre-training step (on side chain conformation prediction) followed by training for $\Delta\Delta G$ prediction. We started from the pre-trained model provided by the authors.

The test Pearson’s correlation results demonstrate that both models suffer from low predictive performance on the small experimental datasets with an effect train-validation test cutoff (Table 1).

Table 1. The performance of $\Delta\Delta G$ prediction methods trained on experimental $\Delta\Delta G$ datasets with a 90% CDR sequence identity train-validation-test cutoff. The models were trained and tested on the ‘Experimental_ $\Delta\Delta G$ _645 – Reverse Mutations + Non-Binders’ (AB-Bind) and on the ‘Experimental_ $\Delta\Delta G$ _608’ (SKEMPI) dataset with 10-fold cross-validation. For the Experimental_ $\Delta\Delta G$ _645 dataset, the correlations are also included for the test dataset excluding the non-binder mutations.

	Test Pearson’s Correlation Experimental_$\Delta\Delta G$_645, 10-fold CV, 90% CDR sequence identity cutoff		Test Pearson’s Correlation Experimental_$\Delta\Delta G$_608, 10-fold CV, 90% CDR sequence identity cutoff
	Incl. non-binders	Excl. non-binders	
DGCddG	0.26	0.32	0.45
RDE-PPI	0.22	0.35	0.48
Graphinity	-0.02	0.19	0.38

The featurization and pre-training strategies appear to give DGCddG and RDE-PPI stronger performance on the small experimental datasets. For RDE-PPI, there may also be some information leakage from the pre-training dataset (PDB-REDO). Neither method, however, is well-suited for scaling to large datasets.

The DGCddG model requires featurization from BLAST and HHblits profiles. We assessed the time required to generate these profiles by applying BLAST and HHblits to two complexes in the synthetic dataset with antigen chains of varying lengths. This took 675 seconds (49 seconds for BLAST and 626 seconds for HHblits across the three WT and one mutant chain; PDB: 7EPX) and 163 seconds (17 seconds for BLAST and 146 seconds for HHblits; PDB: 4OD2). Based on these run times, we estimate that it would take more than 1000 days of compute to create the profiles for the full synthetic dataset, making it infeasible to apply this method.

We were able to retrain the RDE-PPI Network model on the synthetic dataset (Synthetic_FoldX_ΔΔG_942723 dataset, one fold, 90% CDR sequence identity cutoff), starting from the pre-trained model provided by the authors. The final trained model only achieved a test Pearson's correlation of 0.64 (compared to Graphinity's 0.89).

We have updated the paper to include these results.

The tested methods capture the primary strategies used in the field for ΔΔG prediction: featurization (DGCddG), pre-training (RDE-PPI), and end-to-end prediction (Graphinity). We believe this analysis underscores the need for an end-to-end model that does not require time-consuming featurization and that can be readily applied to large-scale datasets. Additionally, the models must be easy to evaluate on datasets with effective train-test cutoffs. The data loading code for RDE-PPI required substantial rewriting to be compatible with non-random data splits, as it was hard-coded for random cross-validation.

We have attempted to make the Graphinity GitHub repository as user-friendly as possible with documentation, examples, and training code so it can be implemented for any ΔΔG prediction task. Additionally, our model takes train, validation, and test files as input to make it easily applicable to non-randomly split datasets.

Updates to the text ("Graphinity performance for predicting experimental ΔΔG"):

For one method, TopNetTree, this leave-one-complex-out test caused a drop in the average Pearson's correlation to 0.17 [17]. For another method, mCSM-AB2, only a minor drop in performance was reported but they appear to include hypothetical reverse mutations in their test data [18]. **Limited performance with train-test cutoffs was also observed for general protein-protein interaction ΔΔG prediction methods which we were able to retrain on the Experimental_ΔΔG_645 dataset (90% CDR sequence identity cutoff); the DGCddG [28] and RDE-PPI Network [29] architectures achieved correlations of 0.26 and 0.22, respectively (Supplementary Table 3, Supplementary Methods).**

Updates to the text ("Using a synthetic dataset of ~1 million mutations"):

Graphinity outperformed other approaches for predicting ΔΔG on this dataset. A simple baseline, the change in number of contacts between the WT and mutant structure (4 Å interaction distance cutoff), achieved a correlation of 0.42 with the synthetic ΔΔG values, less than half the correlation

of our EGNN model. Tree-based (FLAML [30]) and CNN [15] models applied to (structure-informed) sequence inputs achieved similar or lower performance than this baseline (Supplementary Table 3, single fold). When trained on a one-hot encoding of interface residues, these architectures resulted in test Pearson’s correlations of 0.27 and 0.16, respectively. A FLAML model trained on features including the ESM2 [31] embedding of the mutated sequence position and one-hot encoding of the WT and mutant amino acid achieved a correlation of 0.44. **We also assessed further deep learning approaches, including RDE-PPI Network [29], fine-tuned from the pre-trained RDE-PPI Linear, (test Pearson’s correlation 0.64) and another graph-based model built on the Lucidrains implementation of the Equiformer architecture [32, 33]. The Equiformer model achieved similar performance to Graphinity (Pearson’s correlation of 0.89; Supplementary Table 3, single fold), but required substantially more memory and time to train.**

Updates to the text (Supplementary Figures and Tables):

Supplementary Table 3: The performance of $\Delta\Delta G$ prediction methods trained on experimental $\Delta\Delta G$ datasets with a 90% CDR sequence identity train-validation-test cutoff. This analysis includes models which could be retrained with the data splits from this study. The models were trained and tested on the ‘Experimental_ $\Delta\Delta G$ _645 – Reverse Mutations + Non-Binders’ (AB-Bind) and ‘Experimental_ $\Delta\Delta G$ _608’ (SKEMPI) datasets with 10-fold cross-validation. For the Experimental_ $\Delta\Delta G$ _645 dataset, the correlations are also included for the test dataset excluding the non-binder mutations. For more information, see Supplementary Methods.

	Test Pearson’s Correlation		
	Experimental_ $\Delta\Delta G$ _645		Experimental_ $\Delta\Delta G$ _608
	Including non-binders	Excluding non-binders	
DGCddG	0.26	0.32	0.45
RDE-PPI	0.22	0.35	0.48
Graphinity	-0.02	0.19	0.38

Supplementary Table 4: The performance of different models on the Synthetic_FoldX_ $\Delta\Delta G$ _942723 dataset (one fold, held-out test set; 90% length-matched CDR sequence identity cutoff). The same train, validation, and test dataset was used for each model.

Model Input	Model	Test Pearson’s Correlation
One-hot encoded interface residues	FLAML	0.27
	CNN	0.16
ESM2 embedding of mutated position in WT sequence	FLAML	0.44
Embeddings from normalizing flows	RDE-PPI	0.64
Graph of mutation neighborhood	Equiformer	0.89
	Graphinity (EGNN)	0.87

Updates to the text (Supplementary Methods):

Comparison against protein-protein interaction $\Delta\Delta G$ prediction methods

We retrained DGCddG [17] and RDE-PPI Network [18] on the Experimental_ $\Delta\Delta G$ _645 and Experimental_ $\Delta\Delta G$ _608 datasets (90% CDR sequence identity cutoff between train, validation, and test datasets). We installed these methods from GitHub (<https://github.com/lennyiv/DGCddG>, <https://github.com/luost26/RDE-PPI>, respectively) and used training scripts from the repositories (prot-cv/cv_fold_645.py, train_rde_network_skempi.py, respectively) with default parameters. For RDE-PPI, we started training from the pre-trained model provided by the authors (RDE.pt). Additionally, we updated the data loader (rde/dataset/skempi.py) to be compatible with non-randomly split dataset folds. For both methods, the model checkpoint with the lowest validation loss was used for testing.

DGCddG requires featurized inputs generated from BLAST and HHblits profiles. We generated these profiles using PSI-BLAST [19] (version 2.16.0) with the Swiss-Prot database [20] (2025-02-26; as recommended by the authors in private communication), and HHblits (HH-suite3 [21]).

We also trained the RDE-PPI Network method on the synthetic dataset (Synthetic_FoldX_ $\Delta\Delta G$ _942723), using the approach described above. Implementing DGCddG for a dataset of this size was computationally infeasible, however, due to the time required to generate the BLAST and HHblits profiles.

Reviewer #2 (Remarks to the Author):

Hummer, Deane, and colleagues develop an equivariant graph neural network (called 'Graphinity') to predict DDG values for antibody-antigen complexes. The prediction of binding affinity upon antibody sequence change (DDG) is perhaps the biggest open challenge for antibody recognition; there are experimental papers which provide solutions for individual antibodies using massive experimental data, the current state of the art yields overall poor performance for antibodies outside of their immediate training set. The major value - and impact- of this paper is by supplying a lower bound on the amount and diversity of the experimentally data required. This is of considerable importance to the broader field, as it gives the blueprint for machine learning ready data to be generated. The authors first show that the existing highly validated, openly available experimental datasets lead to model overtraining. They then generate synthetic data using the fixed backbone prediction program FoldX and the limited backbone movement Rosetta app 'FlexDDG'. For the FoldX dataset, they sample ~700 mutations each for 1471 antibody-antigen complexes to generate a dataset. They find that their model can recapitulate FoldX performance, while their model shows a comparatively worse performance for FlexDDG, even when controlling for the size of the training set. I have the following small comments.

We thank the reviewer for their thorough reading of our manuscript, clear description of our work, and kind words.

1. For FoldX - the DDG scores are of a different magnitude than the distribution shown in S Fig 1c. Specifically, in Figure 2c, and in Figure 4. The distribution shows a kernel density tailoff between -4 and -6 kcal/mol, while the main text figures show data points out to nearly -100 kcal/mol. Perhaps the discrepancy can be rationalized by the sheer amount of simulated data, such that small numbers of outliers become apparent ? While inclusion of this (small amount of) data should not impact the regression values reported, the results at the tails of these distributions are physically unrealistic. Ideally, these values should be truncated with a softmax or other function before comparing regressions. Alternatively, since this is just synthetic data, a brief description in the text on what are likely non-binding sequences would guide the reader.

We thank the reviewer for this helpful comment and the opportunity for clarification. As the synthetic FoldX dataset was generated through exhaustive mutagenesis of the interface residues, some mutations were predicted that are highly disruptive, would be unlikely to be made experimentally (e.g., Gly to Phe), and, as the reviewer indicates, would abolish binding. Given the nature of the FoldX software, these mutations were assigned very low $\Delta\Delta G$ values, rather than being labeled as non-binders. We decided to include these mutations, given the sandbox nature of this work and for completeness/dataset diversity. However, we agree that it is important to explain that the very low $\Delta\Delta G$ values may not be measurable with experimental techniques (falling below the sensitivity limit). We have now commented on this in the text, using the lowest $\Delta\Delta G$ value in the SKEMPI database (-12.2 kcal/mol) as a reference.

Additionally, we have assessed the model performance with these values removed (test dataset limited to $\Delta\Delta G > -12.2$ kcal/mol). This results in a slight drop in correlation, as the highly disruptive mutations are likely 'easier' to predict than mutations with a small effect on binding affinity.

Updates to the text (“Using a synthetic dataset of ~1 million mutations”):

Additionally, the FoldX dataset includes values for highly disruptive mutations that fall below the range of $\Delta\Delta G$ values that can currently be measured accurately experimentally. While the exact cutoff depends on the sensitivity of the assay and the comparative WT ΔG value, mutations with a $\Delta\Delta G$ value less than -12.2 kcal/mol (the lowest value in the SKEMPI 2.0 database [11]) are likely to be non-binding. Limiting the test dataset to mutations with a FoldX $\Delta\Delta G > -12.2$ kcal/mol (99% of the total values) resulted in a slightly lower test Pearson’s correlation of 0.78.

2. For S. Fig 9, it appears that the model over-performed on mutations to aromatics and underperforms on smaller, polar residues. This suggests that experimental sampling should be biased to mutations to smaller residues. This is important because later in affinity maturation campaigns for antibodies, smaller polar residues often pop up. Perhaps the authors could comment on this finding, or explain why it is not relevant.

We appreciate the reviewer’s insightful suggestion. Mutations to aromatic residues are likely easier to predict, given the potential for a disruptive effect (e.g., clashes). However, it appears the model’s strong performance for these mutations extends further to context sensitivity, as the predictions match the high standard deviation (up to 10 kcal/mol) observed for the true FoldX values.

The average $\Delta\Delta G$ for mutations to small mutations is close to 0 kcal/mol, with standard deviations typically around 1 kcal/mol. The model may be underperforming on these mutations, in part, due to the reduced accuracy of FoldX and the model for mutations with only a small effect on binding affinity.

We agree with the reviewer that it would be valuable to bias sampling toward areas where the model achieves lower performance, and have updated the text to comment on this.

Updates to the text (“Performance by amino acid substitution”):

Certain mutations (e.g., from Gly or to Phe, His, Trp, Tyr) exhibited noticeably lower average $\Delta\Delta G$ values and higher standard deviations, consistent with the disruptive effects that can occur from replacing a small with a large amino acid. **The EGNN model achieved higher performance on these mutations (Supplementary Figure 9). Future experimental data generation should enrich for mutations that proved more challenging to predict (e.g., mutations to small residues).**

3. As a quick technical note - (not requiring author response), there are at least two hypotheses for why Graphinity underperforms on FlexDDG compared with the FoldX datasets, controlling for size of the training set. First, FlexDDG moves the backbone (albeit modestly), and these changes may be harder to capture. Second, for FlexDDG the point of nstruct=35 (rather than nstruct=1) is that the measurement itself is pretty noisy and needs multiple structures to average. Since generating 35 decoys per mutation is prohibitive, there aren’t great rigorous ways that I can think of to quantify this noise.

This is an important point, and we have added a comment in the text.

Updates to the text (“Considerations for generating experimental $\Delta\Delta G$ datasets: dataset size”):
The model was stronger on a FoldX dataset comprised of the equivalent 20,829 mutations (Pearson’s correlation = 0.88; model training for 500 epochs), which may arise from a more complex energy function and/or conformational sampling (including of the backbone) in Flex ddG. Flex ddG is typically implemented by averaging across multiple (default 35) structures, and individual structure models may also be noisy. For the large synthetic dataset generated in this study, computational requirements restricted the number of models we could generate to one per mutation. As a result, there may be a higher level of noise in the Flex ddG values, contributing to the lower predictive accuracy.

4. Supporting Methods. In “Model training times” - should include details for the the Synthetic_FlexDDG dataset.

We apologize for this oversight and have now included the model training times for the synthetic Flex ddG dataset.

Updates to the text (Supplementary Methods, “Model training times”):
Synthetic_FlexddG_ $\Delta\Delta G$ _20829 (500 epochs): ca. 35 hours

Reviewer #2 (Remarks on code availability):

I checked to ensure that the Github was accessible and available, and also looked at the README for Graphinity which contains detailed instructions for installing the application. I also looked at the folders containing the synthetically generated data.

We thank the reviewer for taking the time to check our code and data availability.

Reviewer #3 (Remarks to the Author):

Predicting the change in the binding free energy $\Delta\Delta G$ of antibody-antigen complexes upon mutation is critical in many biological and medical problems. However, related machine learning techniques suffer from the poor availability of experimental data needed for their training and testing. The manuscript introduces an equivariant graph neural network, called Graphinity, to predict $\Delta\Delta G$ from protein structures and tests it on synthetic complexes. The result is that the amount of available data is too small to achieve good performance with this and all competing algorithms; however, Graphinity will potentially outperform the others when a sufficient amount of data is available.

In my opinion, the paper is good and of interest to the community working not only in machine learning but also in antibody design. Therefore, it might deserve publication, although I do have some criticisms, mainly related to the explanation of the results.

We thank the reviewer for their interest in our work, supportive comments, and helpful feedback.

1) In general, I interpret the training and testing of the algorithm on synthetic data generated by FoldX and Rosetta as a sandbox to explore its potential, on the assumption that the data has similar characteristics to real data. For this strategy to work, the data does not have to be indistinguishable from real data, just plausible. Indeed, the $\Delta\Delta G$ predicted by FoldX and Rosetta correlate poorly with experimental values. Nevertheless, these two models are based on a physical description of the interactions that stabilise proteins and, although the parameterisation is far from perfect, they recapitulate the main physical properties of the system. I think this point is not clear from the start and should be emphasised more in the introduction to the manuscript. If one misses this point, one is likely to conclude that the model is not predictive of experimental $\Delta\Delta G$, which (if I understand correctly) is not the goal of this work. The comment of the degree of realism of FoldX (line 157) does not help in this respect. Also (line 186), the further use of Rosetta should not be associated with a more or less realistic dataset, but simply with a different parametrisation, to show that the algorithm is robust if a plausible but different dataset is used.

We thank the reviewer for this feedback on how to more clearly explain the synthetic datasets. We have now updated the text to reflect this. With regards to the text in lines 157 and 186, we felt it would help orient the reader to provide a measure of the correlation between the synthetic and experimental values. However, we agree that this is not the only or absolute metric to assess a synthetic dataset.

Updates to the text ("Introduction"):

In silico prediction of antibody-antigen affinity remains a challenge. Traditional affinity prediction tools, such as FoldX [11] and Rosetta Flex $\Delta\Delta G$ [12], are based on physical equations and empirical measurements. These **methods recapitulate the main physical properties of the system and** have proven effective for certain applications [13] but can be limited in speed and accuracy [12, 14].

Updates to the text ("Using a synthetic dataset of ~1 million mutations"):

We generated a synthetic dataset of nearly 1 million $\Delta\Delta G$ data points (Synthetic_FoldX_ $\Delta\Delta G$ _942723, Supplementary Table 1, Supplementary Figure 1c) by exhaustively mutating the interfaces of structurally-resolved complexes from the Structural Antibody Database (SAbDab)

[24, 25] using FoldX (Figure 1b). **FoldX employs physical equations and empirical measurements to generate predictions of binding affinity. The resulting synthetic dataset will not completely mimic the complexity of true $\Delta\Delta G$ values, but FoldX captures the key features underlying molecular interactions.** The Pearson's correlation between FoldX predictions and experimental values is 0.34 for the AB-Bind dataset [20]. The accuracy is higher for mutations with a larger effect on binding affinity though. The area under the receiver operating characteristic curve (ROC AUC) in predicting whether a mutation is stabilizing or not is 0.87 for mutations with an absolute value greater than 1 kcal/mol [20], **supporting** that this data does contain some of the characteristics of experimental values.

Updates to the text (“Considerations for generating experimental $\Delta\Delta G$ datasets: dataset size”): Additionally, we generated another synthetic dataset using Rosetta Flex ddG [12]. Flex ddG takes substantially longer to run per mutation (Table 1), constraining the number of mutations feasible to model, but offers **an alternative physics-based parameterized method to explore model performance and data requirements. Flex ddG achieved** a higher correlation to experimental values than FoldX, **though both methods recapitulate the physical properties of protein-protein interactions and both are limited in accuracy** (Supplementary Figure 2).

2) I do not understand the decision to assign a $\Delta\Delta G = -8$ kcal/mol to non-binding mutations. Why this? How do the results depend on this choice?

The choice to set the $\Delta\Delta G$ value of non-binding mutations to -8 kcal/mol was made by the authors of AB-Bind and used in this study to enable comparison against previous prediction methods applied to this dataset. The AB-Bind authors included the following comment in their publication: “ $\Delta\Delta G$ was set to be 8 kcal/mol for variants that were experimentally determined to be non-binders, i.e. for variants determined not to bind within the sensitivity of the assay.” (Supporting Information Figure S1 caption; NB AB-Bind uses 8 kcal/mol for Mutant–WT; we implemented the reverse, hence -8 kcal/mol). The authors did not provide further explanation or justification, but we expect they included these mutations due to the sparsity of data, and selected this $\Delta\Delta G$ value as it was at the lower limit of the experimental $\Delta\Delta G$ measurements in the AB-Bind dataset.

The inclusion of the non-binders may impact model performance (Figure 1b) due to the arbitrary assignment of the $\Delta\Delta G$ value. The binding affinity of the non-binders falls below the experimental method sensitivity, however, not all of the corresponding mutations will have an equal effect on binding (some may be more or less disruptive). The EGNN model produces a quantitative prediction and this cannot be meaningfully compared against a binary assignment for a non-binary value.

3) After the thorough analysis in the manuscript, I cannot find a clear statement in the discussion about the minimum size of the experimental dataset needed to get reasonable results in $\Delta\Delta G$ prediction.

We thank the reviewer for raising this point. While we agree that it would be helpful for the community to have an exact aim for dataset size, we felt there were too many contributing factors (e.g., differences between the synthetic and experimental data; the performance metric threshold; the ML architecture used; the dataset diversity; the ability to employ ML and data augmentation strategies to achieve the performance threshold with fewer data points in the future, once there is sufficient signal in the data) to be able to make this statement. Additionally, this value may change over time, with increases in dataset size/information depth and advances in ML. However,

we have now added a sentence to the Discussion to include an estimate under the tested conditions.

Updates to the text ("Discussion"):

We explored the amount of data that would be required for accurate and generalizable prediction of experimental $\Delta\Delta G$ using the synthetic data. Our results with the EGNN and Equiformer architectures, and on the FoldX and Flex $\Delta\Delta G$ datasets suggest that there is currently vastly insufficient experimental data available and orders of magnitude more will likely be needed. **Under the conditions tested in this study, we estimate that at least 90,000 $\Delta\Delta G$ values must be obtained to achieve test Pearson's correlations exceeding 0.85.** Such a volume of data will become more attainable with increases in the throughputs of experimental methods (e.g., [41]). **The exact required dataset size is, however, dependent on dataset diversity and liable to change in the future. As more data becomes available, there is greater** potential for limitations in data to be compensated for, to some extent, by machine learning know-how such as by identifying model architectures that require less data, using stratified sampling or by transfer learning from a related data-rich task or from synthetic data.

4) An interesting point, if easily achievable, would be whether using predicted structures instead of crystallographic ones significantly changes the performance of the predictor, at least within the "sandbox".

We thank the reviewer for this suggestion and have now evaluated the performance of Graphinity on predicted structure inputs. We randomly selected 100 mutations (from the Synthetic_FoldX_ $\Delta\Delta G$ _942723 dataset, 90% CDR sequence identity cutoff, fold 0 test set) and modeled the structures using Boltz-1 (Wohlend et al., 2024; <https://doi.org/10.1101/2024.11.19.624167>). Boltz-1 was run for the WT and mutant complexes, without MSA generation, and excluding mutations that failed due to out-of-memory errors (>300 GB memory).

When Graphinity (without further training) was applied to this dataset of 100 mutations with modeled structure inputs, the test Pearson's correlation was only 0.02 (compared to 0.90 with the solved structure inputs). While this result is not entirely unexpected, it underscores the challenges that remain for modeling the structures of antibody-antigen complexes. This has been seen across the field, for example, with AlphaFold3 reporting 'correct' models for only around 50% of and 'very high accuracy' models for less than 20% of antibody-antigen complexes, even with 10 seeds (Abramson et al., Nature, 2024; <https://doi.org/10.1038/s41586-024-07487-w>; Figure 5a).

This is an important area for further development, both to improve antibody-antigen docking/complex modeling, as well as affinity prediction using predicted inputs.

We have updated the manuscript to comment on these results.

Updates to the text (Using a synthetic dataset of ~1 million mutations):

We also investigated model performance with different graph inputs. **On** the full interface rather than just the mutation site neighborhood, reflecting the input for potential multi-point mutation data, **we** found that performance was maintained (Pearson's correlation = 0.87 on held-out test data, 90% length-matched CDR sequence identity cutoff). **In a preliminary analysis with predicted structure inputs, we applied Graphinity (without further training) to a dataset of 100 randomly selected mutations whose structures we modeled with Boltz-1 [36]. Graphinity was not predictive on this dataset (Pearson's correlation of 0.02), consistent**

with the remaining challenges in modeling high-accuracy structures of antibody-antigen complexes [37]. This is an area where further development and/or specific training will be required.

Updates to the text (Supplementary Methods):

Testing Graphinity on modeled structure inputs

We randomly selected 100 mutations (from the Synthetic_FoldX_ΔΔG_942723 dataset, 90% CDR sequence identity cutoff, fold 0 test set) and modeled the structures using Boltz-1 [22]. Boltz-1 was run for the WT and mutant complexes, without MSA generation, and excluding mutations that failed due to out-of-memory errors (>300 GB memory).

Reviewer #1:

Remarks to the Author:

The reviewer recognized the substantial efforts for the revision.

According to Table1 and S3, Graphinity is almost comparable to the previous PPI prediction model or a bit worse than the existing models. If the reviewer understands the performance correctly, can the author say "Graphinity, an equivariant graph neural network architecture built directly from antibody-antigen structures that achieves state-of-the-art performance on experimental $\Delta\Delta G$ prediction."? If the authors think "our work to provide a lower bound on the amount of data needed, which would frame future method development and data collection efforts", it would be better to emphasize this point in the abstract as well.

We thank the reviewer for raising these points and have updated the abstract and main text accordingly.

Updates to the text (Abstract):

Antibody-antigen binding affinity lies at the heart of therapeutic antibody development: efficacy is guided by specific binding and control of affinity. Here we present Graphinity, an equivariant graph neural network architecture built directly from antibody-antigen structures that **achieves test Pearson's correlations of up to 0.87 on experimental change in binding affinity ($\Delta\Delta G$) prediction**. However, our model, like previous methods, appears to be overtraining on the few hundred experimental data points available. To investigate the amount and type of data required to generalizably predict $\Delta\Delta G$, we built synthetic datasets of nearly 1 million FoldX-generated and >20,000 Rosetta Flex ddG-generated $\Delta\Delta G$ values. Our results indicate there is currently insufficient experimental data to accurately and robustly predict $\Delta\Delta G$, with orders of magnitude more likely needed. Dataset size is not the only consideration – our tests demonstrate the importance of diversity. **These results provide a lower bound on data requirements to inform future method development and data collection efforts.**

Updates to the text (Introduction):

We developed an equivariant graph neural network (EGNN) architecture, Graphinity, which **achieved Pearson's correlations of up to 0.87** on the AB-Bind dataset [20] of 645 single-point mutations.

Reviewer #2:

Remarks to the Author:

The rebuttal and revised manuscript satisfies my minor issues raised.

Reviewer #3:

Remarks to the Author:

The authors addressed all my previous comments. I think the manuscript may be accepted for publication.

We thank the reviewers for their positive feedback and for their comments, which greatly strengthened the manuscript.